**Subject Category:**
Biology (whole organism)

applied mathematics/biomechanics/mechanics

running, stability, rough terrain, uneven terrain, stochastic dynamics, noise propagation

**Author for correspondence:**
Madhusudhan Venkadesan
e-mail: mv@classicalmechanic.net

# Dynamics and stability of running on rough terrains

Nihav Dhawale[1,2], Shreyas Mandre[3]
and Madhusudhan Venkadesan[1]

[1]Department of Mechanical Engineering and Materials Science, Yale University, New Haven, CT 06520, USA
[2]National Centre for Biological Sciences, Tata Institute of Fundamental Research, Bangalore, Karnataka 560065, India
[3]School of Engineering, Brown University, Providence, RI 02912, USA

 SM, 0000-0002-1525-8325; MV, 0000-0001-5754-7478

Stability of running on rough terrain depends on the propagation of perturbations due to the ground. We consider stability within the sagittal plane and model the dynamics of running as a two-dimensional body with alternating aerial and stance phases. Stance is modelled as a passive, impulsive collision followed by an active, impulsive push-off that compensates for collisional losses. Such a runner has infinitely many strategies to maintain periodic gaits on flat ground. However, these strategies differ in how perturbations due to terrain unevenness are propagated. Instabilities manifest as tumbling (orientational instability) or failing to maintain a steady speed (translational instability). We find that open-loop strategies that avoid sensory feedback are sufficient to maintain stability on step-like terrains with piecewise flat surfaces that randomly vary in height. However, these open-loop runners lose orientational stability on rough terrains whose slope also varies randomly. The orientational instability is significantly mitigated by minimizing the tangential collision, which typically requires sensory information and anticipatory strategies such as leg retraction. By analysing the propagation of perturbations, we derive a single dimensionless parameter that governs stability. This parameter provides guidelines for the design and control of both biological and robotic runners.

## 1. Introduction

Legged terrestrial animals run stably on rough terrains, despite potential difficulties such as sensory latencies and the highly dynamic nature of running. Our current understanding of how running animals negotiate rough terrains is based on studies where the animal experiences obstacles in the form of a single step up or down [1,2], or a random sequence of up and down steps [3,4]. However, natural terrains exhibit not only variations

in height but also in slope, and it is unclear how our understanding of running on step-like terrains translates to such natural terrains.

Mathematical studies of running over rough terrains reflect the experiments and focus on stability when running on step-like terrains that are piecewise flat [5–7]. Furthermore, models of runners with massless legs, such as the spring-legged-inverted-pendulum (SLIP) [8–10], cannot distinguish between different slopes of the terrain and only respond to variations in height. Assuming a massless leg enforces the ground reaction force vector to always align with the leg [11], regardless of the terrain's slope beneath the foot. More detailed models that mimic the anatomy of specific animals or robots avoid this limitation of SLIP models [7], at the cost of generalizability. Thus, there is a need for generalizable models of running that incorporate dependence on both terrain slope and height, and which remain sufficiently abstract to glean principles that may underlie stability on rough terrains.

Stability may be governed by many factors, including sensory feedback control [12–14], the inherently stabilizing mechanical response of the animal's body [15], energy dissipation within the body [1], and feed-forward strategies such as swing-leg retraction [16]. The slowest is often sensory feedback control that has latencies comparable to or greater than the stance duration. For example, at endurance running speeds for humans, the stance lasts around 200 ms [17] and only slightly longer than the shortest proprioceptive feedback delay of 70–100 ms or visual feedback delay of 150–200 ms [18]. To better understand the inherent stability or instability of the dynamics of running, we consider only passive mechanical and anticipatory strategies in this study without relying on active feedback control.

Studies of running birds and the role of open-loop stability of running find that increased energy dissipation during stance may help stability when faced with an unexpected drop in terrain height [1]. Consistent with the role of energy dissipation, experiments with humans find that metabolic power increases by 5% to run on step-like terrains versus flat ground [4]. Walking over rough terrains leads to an increase of 28% in metabolic power [19], higher in both relative and absolute terms. The difference in energetics may indicate that the dynamics of running are inherently less unstable, but such an analysis on natural rough terrains has not been carried out. Therefore, we incorporate energy dissipation in our examination of open-loop strategies to address the effect of dissipation on stability.

Not relying on feedback control within a single stance does not preclude active strategies that rely on anticipation or internal models, sometimes called feed-forward strategies. Computational studies of walking have demonstrated the role of look-ahead strategies that use the height and slope of the oncoming terrain in planning the control [20]. Evidence for the importance of feed-forward strategies for running come from computational studies of SLIP-like running dynamics [16] that show how swing-leg retraction automatically modulates the landing angle in response to unexpected variations in the terrain height. However, these studies on running have not yet considered the effect of slope variations in the terrain. Thus, in our study, we analyse anticipatory strategies that incorporate the slope of the oncoming terrain.

An extreme and simplified approximation of running is that of a point mass with an impulsive and instantaneous stance followed by projectile flight. Such an approximation appears as a natural solution to the problem of minimizing measures of metabolic energy consumption when the desired forward speed exceeds critical levels and subject to other constraints such as step length [21]. SLIP-like models are an unfolding of these point-mass instantaneous-stance models to have finite stance duration. They have helped us understand the kinetics of stance [15], the energetics of producing forces on flat terrains [21], and the role of swing-leg retraction on piecewise flat terrains [6]. However, these point-mass models possess no sense of body orientation during the aerial phase and are therefore immune to falling by tumbling.

In this study, we unfold the point-mass, instantaneous-stance model by using a finite moment of inertia for the runner, while still maintaining an impulsive stance. A finite moment of inertia defines a body orientation and thus enables the examination of the effect of angular momentum fluctuations induced by stance. Such a model maps the net effect of the ground forces over stance as a linear impulse applied at the contact point and an angular impulse applied at the centre of mass. These impulses lead to a change in the linear and angular momentum of the whole runner because of the passive and active forces during stance. As we discuss later, the angular impulse captures the effect of a finite stance duration and configuration changes during stance.

Section 2 develops a sagittal-plane model that incorporates a finite moment of inertia, inelastic two-dimensional collisions and an active push-off so that both terrain slope and height variations affect stability. In §3, we use Monte Carlo simulations with random variation of ground height and slope to examine open-loop strategies, the effects of energy dissipation, and strategies that anticipate the

slope of the terrain. Section 4 derives the linearized dynamical equations and analyses their stability. Using the linearization, we find a single dimensionless parameter that governs stability in §5, which in turn guides morphological design for stability. We conclude in §6 with a discussion of the limitations and generality of our analyses, its relationship to experimental results and generate testable predictions for future experiments.

# 2. Mathematical model of sagittal-plane running

We model the runner in the sagittal plane as a disc-like rigid body (figure 1*a,b*) of mass $m$, radius of gyration $r_g$, i.e. moment of inertia $I_{/G} = mr_g^2$ about its centre of mass, and radius $r_\ell$ (leg length). All quantities are in units such that $m = 1$, $r_\ell = 1$ and the acceleration due to gravity $g = 1$. See §7 for notation used in this paper.

## 2.1. Aerial and stance phases

A single step comprises an aerial and a stance phase. The aerial phase is modelled as a drag-free projectile in uniform gravity. Stance involves two successive parts: a passive collision with the ground followed by an active push-off. The passive collision is two-dimensional and parametrized by two coefficients of restitution: $\epsilon_n$ along the normal to the ground and $\epsilon_t$ along the tangent to the ground. The active push-off applies a linear impulse $J_{imp}$ at the contact point P and a rotational impulse $J_\phi$ at the centre of mass G. The governing dynamical equations are

$$\text{passive collision:} \quad v_{P,t}^c = \epsilon_t v_{P,t}^-, \quad v_{P,n}^c = -\epsilon_n v_{P,n}^- , \tag{2.1a}$$

$$H_{/P}^c - H_P^- = 0, \tag{2.1b}$$

$$\text{push-off:} \quad \boldsymbol{v}_G^+ = \boldsymbol{v}_G^c + \boldsymbol{J}_{imp}, \quad I_{/G}\omega^+ = I_{/G}\omega^c + J_{imp,t} + J_\phi, \tag{2.1c}$$

$$\text{flight:} \quad \ddot{x}_G(t) = 0, \quad \ddot{y}_G(t) = -1, \quad \ddot{\phi}(t) = 0 \tag{2.1d}$$

$$\text{and initial conditions:} \quad \begin{pmatrix} x_G(0) \\ y_G(0) \\ \phi(0) \end{pmatrix} = \begin{pmatrix} x_G^+ \\ y_G^+ \\ \phi^+ \end{pmatrix}, \begin{pmatrix} \dot{x}_G(0) \\ \dot{y}_G(0) \\ \dot{\phi}(0) \end{pmatrix} = \begin{pmatrix} v_{G,x}^+ \\ v_{G,y}^+ \\ \omega^+ \end{pmatrix}. \tag{2.1e}$$

Horizontal and vertical positions are denoted by $x$ and $y$, respectively, orientation by $\phi$, velocity by $v$, angular velocity by $\omega$, and angular momentum by $H$. Subscripts of 't' and 'n' refer to the tangential and normal directions to the terrain, while subscripts of 'x' and 'y' refer to the horizontal and vertical directions in the laboratory frame (figure 1*b*). Superscript '−' denotes variables immediately preceding the collision, '*c*' after the passive collision and '+' after the active push-off. Subscripts P and G refer to quantities associated with the contact point and centre of mass, respectively.

The mechanical state of the runner is parametrized by the centre of mass positions $(x_G, y_G)$, body orientation $\phi$ and their respective velocities $(v_{G,x}, v_{G,y})$ and $\omega$. Because the stance is assumed to be instantaneous, the velocities may change discontinuously but the position and orientation remain constant during stance. The instantaneous stance assumption also implies that finite forces such as gravity or air-drag do not contribute to the impulse on the runner. However, the active rotational impulse $J_\phi$ applied at the centre of mass G captures the effects of varying posture over stance and the changing centre of pressure on the ground. We examine this approximation and its implications in the discussion.

## 2.2. Stance: passive collision

The passive collision has two components, the normal and the tangential, which are parametrized by $\epsilon_n$ and $\epsilon_t$, respectively. Modulating these coefficients of restitution would exert control over the passive collision. We hold these parameters constant from step to step and examine the effect of different values on stability.

An animal may vary the extent to which the normal momentum impinging on the terrain is stored elastically and later used for push-off. Without detailed consideration of how an animal may accomplish this, we allow our model to precisely specify $\epsilon_n$. The normal collision is perfectly elastic for $\epsilon_n = 1$ and perfectly inelastic for $\epsilon_n = 0$.

The tangential collision directly affects the body's angular momentum. We, therefore, elaborate the model to consider different ways in which this collision may be controlled. One strategy for an animal

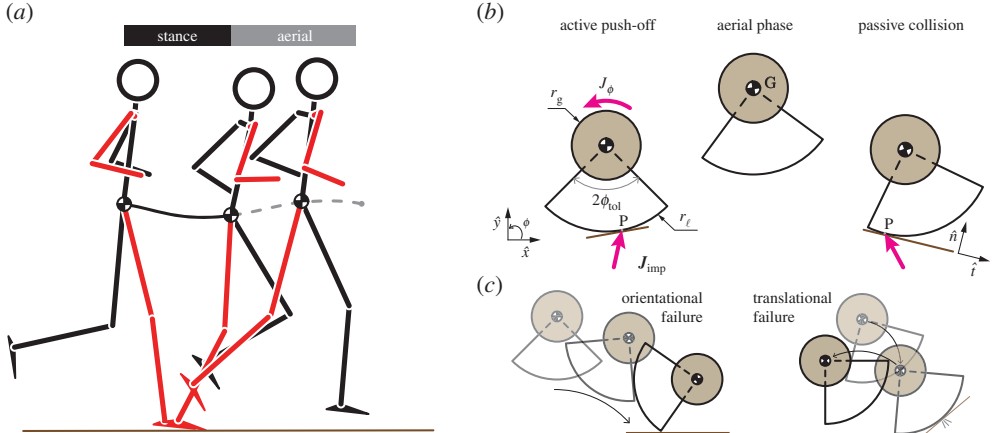

**Figure 1.** Bouncing as a model of running. (*a*) The outline of a human running at 3.5 m s$^{-1}$, created from motion capture data, shows stance and aerial phases over a single step. The stance leg and ipsilateral arm are in red, and the centre of mass trajectory is shown as a dashed curve during flight and solid curve during stance. (*b*) The runner pushes-off the ground by applying a linear impulse $\boldsymbol{J}_{\text{imp}}$ at the contact point P, and the effect of additional torques about the centre of mass arising from configuration changes during stance are captured by an angular impulse $J_\phi$ at the centre of mass. At the end of the aerial phase, the runner undergoes a passive collision with the ground at the new contact point P. The momentum lost due to the collision in directions tangential and normal to the terrain surface is dictated by the parameters $\epsilon_t$ and $\epsilon_n$, respectively. (*c*) The runner can fail in two ways: *orientational failure* when orientation at touchdown exceeds the tip-over threshold, i.e. $|\phi^-| > \phi_{\text{tol}}$, or *translational failure* when the forward velocity at take-off drops below a chosen threshold, e.g. $v_{G,x}^+ < 0.01$.

to modulate the tangential collision is to vary the foot speed along the terrain's tangential direction. For the same speed of the centre of mass, the foot speed may be varied through actions such as retracting the leg. However, the control of the foot's speed relative to the terrain's tangential direction requires sensory information on the ground speed and the terrain's slope. In our model, this corresponds to knowledge of $v_{P,t}^-$, the speed of the contact point along the terrain's tangent in the moment just before the collision. We consider two extremes for the availability of such sensory information. At one extreme is an *open-loop strategy* that has no information and assumes $v_{P,t}^- = v_{x0}$, which implies three assumptions: forward speed equals the initial speed, the terrain is flat and the body has no spin. In this case, the intended tangential coefficient of restitution $\epsilon_{tc}$ and the actual coefficient $\epsilon_t$ would generally not equal each other and depend on the ratio $v_{x0}/v_{P,t}^-$. At the other extreme is an *anticipatory strategy* with perfect *a priori* information so that $\epsilon_{tc} = \epsilon_t$. These are summarized as

$$\epsilon_t = \begin{cases} \epsilon_{tc} \dfrac{v_{x0}}{v_{P,t}^-} & : \text{open-loop,} \\ \epsilon_{tc} & : \text{anticipatory.} \end{cases} \tag{2.2}$$

We derive this equation (2.2) more formally in electronic supplementary material, S3.2 and examine the relationship between $\epsilon_{tc}$ and $\epsilon_t$ for the open-loop strategy in electronic supplementary material, S4.1.

## 2.3. Stance: active push-off

Stance ends with the application of an active, linear push-off impulse $\boldsymbol{J}_{\text{imp}}$ at the contact point P and an active angular push-off impulse $J_\phi$ at the centre of mass G. We constrain these impulses so that in the absence of external perturbations or other disturbances, the runner is perfectly periodic and remains upright ($\phi(t) = 0$) on flat ground. Importantly, once the impulses are chosen for flat ground, they are not allowed to vary step to step on any other terrain to reflect the absence of active feedback control. Together, these conditions imply that the active push-off impulses $\boldsymbol{J}_{\text{imp}}$ and $J_\phi$ depend only on $\epsilon_n$, $\epsilon_{tc}$, $v_{x0}$, $v_{y0}$ and $I_{/G}$, and no other parameters, according to

$$\boldsymbol{J}_{\text{imp}} = \begin{pmatrix} v_{x0} \\ v_{y0} \end{pmatrix} - \begin{pmatrix} \left(\epsilon_{tc} + \dfrac{1-\epsilon_{tc}}{1+I_{/G}}\right) v_{x0} \\ -\epsilon_n v_{y0} \end{pmatrix} \tag{2.3a}$$

and

$$J_\phi = 0. \tag{2.3b}$$

**Table 1.** Parameter values for a human-like runner. Units are chosen such that $m = 1$, $g = 1$ and $r_\ell = 1$. Parameters describing the runner are discussed in §2. The heights $h$ at grid points defining the terrain are chosen from a uniform distribution over the range $[-0.03, 0.03]$ (electronic supplementary material, S1.3). The ensemble size used in the Monte Carlo simulations is $M$, and MAX is the number of steps to which the runner is simulated (electronic supplementary material, S2). All runners failed before reaching MAX. We elaborate on the choice of these values in electronic supplementary material, S1.2.

| runner | | | | | terrain | | Monte Carlo | |
|---|---|---|---|---|---|---|---|---|
| $l_{/G}$ | $\epsilon_n$ | $\phi_{tol}$ | $v_{x0}$ | $v_{y0}$ | $h$ | $\lambda$ | $M$ | MAX |
| 0.17 | 0.63 | $\pi/6$ | 0.96 | 0.26 | $\sim \mathcal{U}(-0.03, 0.03)$ | 0.1 | $10^5$ | $10^3$ |

Thus, the centre of mass of a periodic runner on flat ground has a constant forward speed $v_{x0}$ and vertical speed $v_{y0}$ at every step.

On rough terrains, there are two options for defining the application of the invariant linear impulse on every step. First, the impulse vector $J_{imp}$ may be held constant in every step with respect to gravity ($\hat{x} - \hat{y}$ frame in figure 1$b$), which we call the *laboratory-fixed* push-off policy. Second, the impulse vector may be held constant in every step with respect to the normal direction to the terrain at the point of contact ($\hat{t} - \hat{n}$ frame in figure 1$b$), which we call a *terrain-fixed* push-off policy. The terrain-fixed policy may be considered a better approximation of what animals do, because the normal to the terrain and the leg orientation are often coupled, whereas leg orientation at contact and gravity may vary from step to step. Implicit in preferring the terrain-fixed policy is the assumption that joint torques to apply forces are planned in an ego-centric (body-fixed) frame of reference. For the disc-like model of a runner that we use, the terrain-fixed and body-fixed policies are identical. Detailed expressions for the velocities in the stance phase, as well as expressions for $J_{imp}$ under both push-off policies are given in electronic supplementary material, S3. We present a complete analysis of the laboratory-fixed push-off policy in electronic supplementary material, S10 and focus on the terrain-fixed policy in the main paper. The active linear and angular impulses were found for a steady, periodic gait on flat terrain and held fixed thereafter. For a steady gait, the angular impulse $J_\phi = 0$ at all times, whereas the linear impulse depends on the collision parameters and speed. Section 6 examines the role of $J_\phi$ in extending the model to account for finite stance durations.

# 3. Monte Carlo simulations

A sagittal-plane runner can only fail by two modes, when the body orientation exceeds a chosen threshold (*orientational failure*) or by failing to move forward any longer (*translational failure*). We choose the orientational threshold $\phi_{tol}$ as the angle of tilt to passively topple a human who is standing with their feet apart in a pose resembling double-stance in walking.

We perform Monte Carlo simulations on step-like terrains and undulating rough terrains to estimate the statistics of failure for both open-loop and anticipatory runners. These simulations use different combinations of the normal and tangential collision parameters, $\epsilon_n$ and $\epsilon_{tc}$, respectively. Stability is quantified by the mean steps to failure, like previous studies of rough terrain walking [20].

The terrain is modelled as a piecewise linear interpolation of an underlying random grid. The grid points are separated by a distance $\lambda$ and the heights $h$ of the grid points are chosen from a uniform random distribution (table 1; electronic supplementary material, S1.3). A linear interpolation between the grid points yields a terrain with random variations in both slope and height. Corners at grid points imply an indeterminate slope, and we, therefore, define an effective slope at the grid points by interpolating the slope before and after the point (details in electronic supplementary material, S1.4). Parameter values that represent a human-like runner (table 1) are used for all Monte Carlo simulations, unless indicated otherwise.

## 3.1. Open-loop runners on rough terrains

Open-loop runners always fail through an orientational instability on rough terrains regardless of the energy dissipated per step (figure 2$a$,$d$). The open-loop runners with human-like inertia and size take $9.6 \pm 4.1$ steps (mean $\pm$ s.d.) before tumbling while only 1% of the runners fall within three steps (figure 2$a$). Decreasing $\epsilon_n$

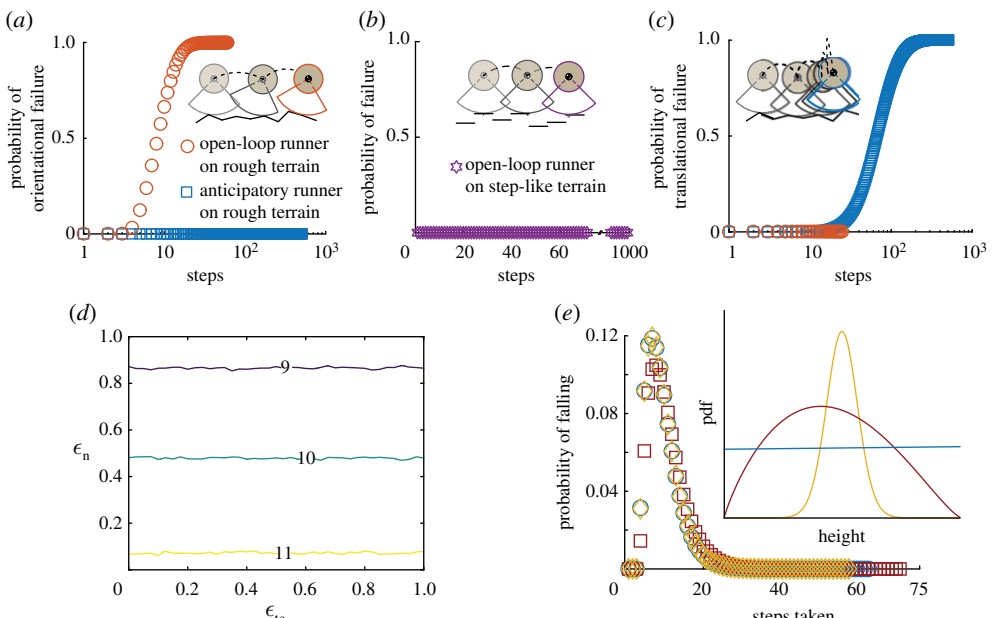

**Figure 2.** The effect of the tangential collision, energy dissipation and terrain geometry on running stability for a human-like runner found using Monte Carlo simulations. (*a*) Open-loop runners with $\epsilon_{tc} = 0$ (orange circles) lose orientational stability on the rough terrain, while anticipatory runners with $\epsilon_{tc} = 1$ (blue squares) maintain orientation. (*b*) On the step-like terrain, open-loop runners (purple stars) maintain forward speed and orientation as the probability of failure, orientational or translational is zero. Open-loop and anticipatory runners are identical on step-like terrains. (*c*) Anticipatory runners slow down on the rough terrain, eventually completely losing forward speed. Whereas human-like open-loop runners also lose forward speed, they lose orientational stability before completely losing forward momentum. (*d*) A contour plot of mean steps taken by open-loop runners as a function of $\epsilon_{tc}$ and $\epsilon_n$ finds that contours are approximately parallel to the $\epsilon_{tc}$ axis, while the steps taken increases with decreasing $\epsilon_n$. (*e*) Steps to failure distributions for human-like open-loop runners on rough terrain with height distributions for the grid points drawn from von Mises (yellow diamonds, mean = 0, $\kappa = 6$), beta (dark red squares, $\alpha = 1.9$, $\beta = 2.3$) and uniform distributions (blue circles). The inset shows the three different probability density functions used to generate the terrain: von Mises (yellow), beta (dark red) and uniform (blue). The distributions were scaled and shifted such that mean height $= 0$ and range $= 0.060 r_\ell$ (table 1; electronic supplementary material, S1.2).

from 1 to 0 increases the mean steps to failure by just two steps (figure 2*d*). Thus, dissipating more energy per step in the normal collision has minimal influence on stability. The tangential collision, parametrized by $\epsilon_{tc}$, has little or no influence because the contour lines of the mean steps to failure are nearly parallel to the $\epsilon_{tc}$ axis (figure 2*d*). Therefore, energy dissipation or modulating the tangential collision are both ineffective stabilization strategies for purely open-loop runners.

## 3.2. Open-loop runners on step-like terrains

The purely open-loop runner remains stable on step-like terrains that are piecewise flat and possess only height variations (figure 2*b*). This is because forward and vertical dynamics are decoupled on piecewise flat terrains, and hence, the open-loop runner does not fall as long as the step height is smaller than the apex height of the aerial phase. This result suggests a foot placement strategy for running on any rough terrain, namely, to aim to land on flat patches of the ground so that stability is maintained with little reliance on feedback control. However, such a strategy would require visual surveying of the terrain up ahead and planning the location of foot falls. Future experiments that track foot placement on rough terrains could test whether a runner uses a strategy of aiming for flat regions of the terrain by estimating the fraction of footfalls on flat versus highly sloped regions.

## 3.3. Effect of terrain geometry

The exact step-to-step variation in the terrain's height and slope depend on the distribution function used to generate the random terrain (figure 2*e*, inset). However, we find that the distribution underlying the rough terrain has little effect on the distribution of the steps to failure (figure 2*e*) when assessed using

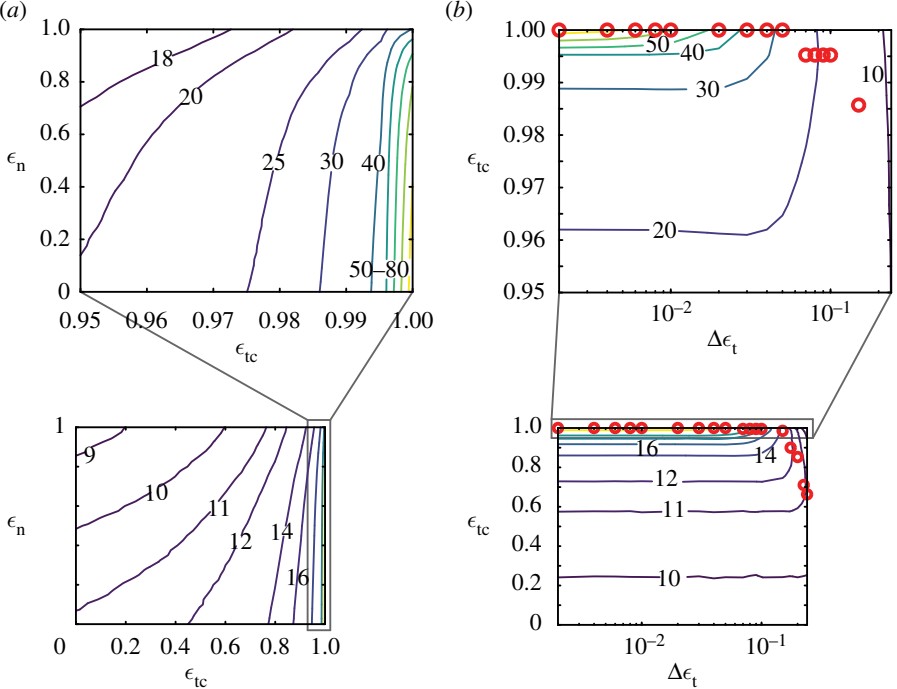

**Figure 3.** Effect of tangential collisions and energy dissipation on running stability for anticipatory runners. In each panel, the contour plot of mean steps taken over the entire range of independent parameters is shown together with a zoom-in of contours that are bunched together. (a) The contour plot of mean steps taken by anticipatory runners as a function of $\epsilon_{tc}$ and $\epsilon_n$ shows that contours are bunched close together around $\epsilon_{tc} \simeq 1$, with the maximum steps taken at $\epsilon_n = 0$, $\epsilon_{tc} = 1$ (top plot) and minimum at $\epsilon_n = 1$, $\epsilon_{tc} = 0$ (bottom plot). (b) Effect of noise in $\epsilon_{tc}$. Contour plot of mean steps taken as a function of $\epsilon_{tc}$ and $\Delta\epsilon_t$. The optimal $\epsilon_{tc}$ (red circles) is shown for each value of $\Delta\epsilon_t$ simulated.

three different functions to generate rough terrains: von Mises, uniform and beta. Runners took $9.6 \pm 4.1$ (mean $\pm$ s.d.) steps before tumbling on terrains characterized by von Mises and uniform distributions, and $10.8 \pm 4.7$ steps on the terrains characterized by the beta distribution. All steps to failure distributions are unimodal, but skewed. A Markov model for the step-to-step dynamics (electronic supplementary material, S8) lends insight into the nearly invariant shape of the steps-to-failure distribution. The insensitivity may arise from the terrain roughness being uncorrelated from step to step (terrain's correlation length $\lambda \ll 1$), and thus the net effect of the perturbations resembles a Gaussian noise process that is propagated by the dynamics of running.

## 3.4. Anticipatory runners on rough terrains: tangential collisions

Runners that use anticipatory strategies to control the tangential passive collision maintain orientational stability if they entirely avoid tangential collisions using $\epsilon_{tc} = 1$. But these runners eventually fail by completely losing forward momentum (figure 2c). Recall that because of the active push-off, the loss of forward momentum is not simply a break of symmetry by the passive tangential collision. Through a more careful analysis, we find that the mean slope encountered by the runners is positive and not zero, i.e. the terrain preferentially impedes the forward momentum (electronic supplementary material, figure S10d). For human-like parameters, anticipatory runners ($\epsilon_{tc} = 1$) take $75 \pm 40.9$ (mean $\pm$ s.d.) steps before completely losing forward momentum and only 1% of the runners stop moving forward within 15 steps (figure 2c). By contrast, over 80% of the open-loop runners fail within 15 steps.

For the anticipatory runner, permitting tangential collisions $\epsilon_{tc} < 1$ induces orientational failures and the mean steps to failure decreases. For example, with $\epsilon_n = 0$, the mean steps to failure when $\epsilon_{tc} = 1$ is 85 and decreases to 20 when $\epsilon_{tc} = 0.95$ (figure 3a). A 5% decrease in $\epsilon_{tc}$ caused an over threefold decrease in the mean steps to failure. Importantly, the dominant mode of failure switches from translational failures to orientational failures (electronic supplementary material, figure S5a). At $\epsilon_{tc} = 0$ and independent of $\epsilon_n$, the anticipatory and open-loop strategies are identical. Thus, the anticipatory runner substantially improves stability by avoiding tangential collisions.

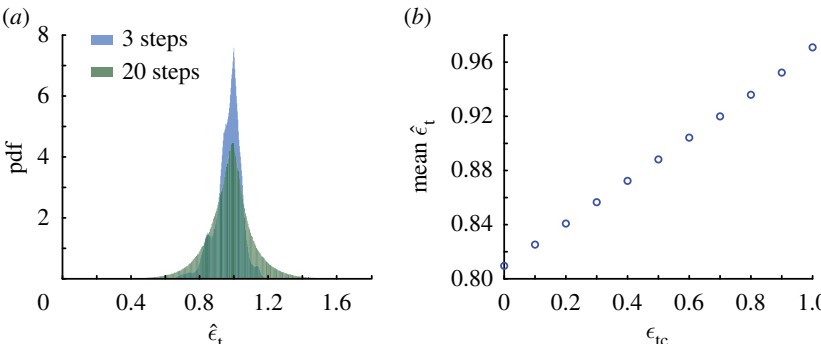

**Figure 4.** Estimated tangential coefficient of restitution $\hat{\epsilon}_t$ for anticipatory runners using Monte Carlo simulations with an ensemble size of $10^6$. (*a*) Probability density function of $\hat{\epsilon}_t$ for human-like anticipatory runners with $\epsilon_{tc} = 1$ on rough terrain after three steps and after 20 steps. While the standard deviation almost doubles between the two distributions shown here (electronic supplementary material, figure S6*c*); the mean of the distribution converges by three steps (electronic supplementary material, figure S6*b*). (*b*) Mean $\hat{\epsilon}_t$, converges by three steps for all values of $\epsilon_{tc}$ (electronic supplementary material, figure S6*b*) and is always less than 1, ranging from 0.81 at $\epsilon_{tc} = 0$ to 0.97 at $\epsilon_{tc} = 1$.

Increasing energy dissipation in the normal collision increases the number of steps taken by the anticipatory runner. For example, at $\epsilon_{tc} = 1$, where runners only fail by losing forward speed, increasing energy dissipation by changing from $\epsilon_n = 1$ to $\epsilon_n = 0$, increases the mean number of steps taken before failure by twofold, from 40 to 85 (figure 3*a*). Away from $\epsilon_{tc} = 1$, energy dissipation has a smaller effect on stability. When $\epsilon_{tc} \approx 0$, the anticipatory runners resemble the open-loop runners and the mean steps to failure increases by only two steps despite $\epsilon_n$ decreasing from 1 to 0 (figure 3*a*). Thus, for the anticipatory runner using $\epsilon_{tc} \approx 1$, increasing energy dissipation in the direction normal to the terrain is an effective means to improve stability, unlike for the open-loop runner.

## 3.5. Noise in anticipatory strategies

The sensitivity of the steps to failure with respect to tangential collisions prompts an examination of the effect of stochasticity in how a runner may control the tangential collision. After all, no runner can exactly control the tangential collision from step to step. For example, errors in sensing the terrain profile as well as motor noise may prevent accurate implementation of a desired $\epsilon_{tc}$. We model such sources of noise in controlling the tangential collision as

$$\epsilon_{t,noisy} = \epsilon_{tc} + \Delta\epsilon_t \eta, \tag{3.1a}$$

$$\text{where } \eta \sim \mathcal{U}[-1, 1], \quad \Delta\epsilon_t \in \mathbb{R}. \tag{3.1b}$$

The uniformly distributed zero-mean random variable $\eta$ models random step-to-step noise in $\epsilon_{tc}$ and $\Delta\epsilon_t$ is the noise intensity. The noisy $\epsilon_{t,noisy}$ is allowed to exceed 1 in our simulations.

We find that incurring tangential collisions ($\epsilon_{tc} < 1$) is optimal when there is non-zero noise ($\Delta\epsilon_t > 0$). This is unlike the noiseless anticipatory runner whose optimum is $\epsilon_{tc} = 1$. Moreover, noise in controlling tangential collisions does affect stability and the mean steps to failure are severely reduced (figure 3*b*). For example, compared to a noiseless human-like runner, the mean steps to failure drops ninefold for a runner with noise intensity $\Delta\epsilon_t = 0.1$, and the optimum $\epsilon_{tc}$ decreases by 1% to $\epsilon_{tc} = 0.99$ (figure 3*b*). Additional noise in the tangential collision of open-loop runners reduces the number of steps taken, but does not alter the dependence of steps taken on $\epsilon_{tc}$ (electronic supplementary material, S4.2). Therefore, for anticipatory runners, noise in controlling the tangential collision implies that incurring a slight tangential collision is optimal but at the cost of stability.

## 3.6. Predictions for $\epsilon_t$ in experiments

A main finding of our analyses is the importance of minimizing tangential collisions with the ground when running on rough terrains. But measuring $\epsilon_t$ on rough terrains is challenging because it needs a well-defined point of contact under the foot, precise knowledge of the terrain's slope in three dimensions at that point and measurement of the reaction force along that tangent. To facilitate

comparisons with experimental data, we consider an easier way to measure $\epsilon_t$ via the parameter $\hat{\epsilon}_t$ that is defined as

$$\hat{\epsilon}_t = 1 - \frac{\Delta v_x}{v_x}, \tag{3.2}$$

where $\Delta v_x / v_x$ is the fraction of the forward momentum of the runner lost due to the passive collision. On perfectly flat terrain, $\epsilon_t = \hat{\epsilon}_t$.

In the Monte Carlo simulations, $\hat{\epsilon}_t$ is characterized by a distribution that evolves with increasing steps (figure 4a; electronic supplementary material, figure S6a). The dependence of $\hat{\epsilon}_t$ on steps taken arises because the runner is slowing down, and thus $v_x$ and consequently $\Delta v_x$ change from step to step. But, the mean of $\hat{\epsilon}_t$ appears to converge to a constant after just three steps for all values of $\epsilon_{tc}$ (electronic supplementary material, figure S6b). Importantly, mean $\hat{\epsilon}_t$ increases linearly with $\epsilon_{tc}$ (figure 4b) and is thus a reliable correlate of the true tangential collision. However, $\hat{\epsilon}_t$ has a reduced range; mean $\hat{\epsilon}_t = 0.81$ at $\epsilon_{tc} = 0$, and mean $\hat{\epsilon}_t = 0.97$ at $\epsilon_{tc} = 1$. The standard deviation of the distributions converges to a value between 0.05 and 0.1 by approximately 10 steps for most values of $\epsilon_{tc}$ except when $\epsilon_{tc} \to 1$ (figure 4a; electronic supplementary material, figure S6b). For comparison, reported values of $\hat{\epsilon}_t$ from experiments with human runners on flat and two rough terrains are $0.94 \pm 0.01$ (mean $\pm$ s.d.) identically [22]. These experimental data are consistent with the prediction that optimal anticipatory runners should maintain $\epsilon_{tc} = 1$.

# 4. Linear stability analysis

For periodic dynamic systems, linear stability is defined as the response to small perturbations in the neighbourhood of a periodic orbit [15,23,24] and analysed using the Floquet theory [15,25]. Floquet analysis for the stability of a periodic orbit defines a transverse cross-section to the orbit and a discrete return map from initial conditions on the cross-section back to the same cross-section after a complete period. The eigenvalues of the return map, called Floquet multipliers, are independent of the chosen cross-section and govern the stability of the periodic solution to small perturbations [25]. Here, we consider the anticipatory runner and discuss the open-loop runner in electronic supplementary material, S6 because the unstable modes of both variants are the same.

The mechanical state of the runner is represented by $\boldsymbol{\zeta} = (x, y, \phi, v_x, v_y, \omega)^T$, where $(x, y)$ and $\phi$ denote the centre of mass position and orientation, and $(v_x, v_y)$ and $\omega$ are the respective velocities, all measured in a Newtonian reference frame that translates forward at a constant speed $v_{x0}$. A steady runner is periodic in this translating Newtonian frame of reference. We define a transverse cross-section (Poincaré section) at the apex of the aerial phase ($v_y = 0$), following the approach of Full et al. [23] and Seyfarth et al. [16]. Equations (2.1) yield the step-to-step return map $f_{an}$ and its linearization $T_{an}$ in terms of the mechanical state $\boldsymbol{\psi}$ in a translating frame according to

$$\boldsymbol{\psi} = (x, y, \phi, v_x, \omega)^T, \tag{4.1a}$$

$$\boldsymbol{\psi}_{n+1} = f_{an}(\boldsymbol{\psi}_n), \tag{4.1b}$$

$$\Delta\boldsymbol{\psi}_{n+1} = T_{an}\Delta\boldsymbol{\psi}_n, \tag{4.1c}$$

$$\text{where } \Delta\boldsymbol{\psi} = \boldsymbol{\psi} - \boldsymbol{\psi}^*, \quad T_{an} = \frac{\partial f_{an}}{\partial \boldsymbol{\psi}}\Big|_{\boldsymbol{\psi}^*}. \tag{4.1d}$$

The Poincaré map given by equation (4.1b) has a fixed point at $\boldsymbol{\psi}^* = 0$ when the terrain is flat and corresponds to an exactly periodic runner on flat ground (figure 5).

The linearized return map $T_{an}$ has three eigenvalues equal to 1 and the others are all less than 1. The eigenvalues with magnitude less than 1 correspond to stable modes so that perturbations along their respective eigenvectors will always decay. The remaining three eigenvalues are all $\lambda = 1$ with algebraic multiplicity equal to 3 and geometric multiplicity equal to 2. This implies that there are only two independent eigenvectors corresponding to the three unity eigenvalues and matrix $T_{an}$ is, therefore, non-diagonalizable. For non-diagonalizable systems, the Jordan decomposition is used to analyse stability in terms of generalized eigenvectors (electronic supplementary material, S6) and implies that the modes (eigenvectors) associated with these eigenvalues cannot be decoupled and analysed independently.

The two eigenvectors $\boldsymbol{v}_1$ and $\boldsymbol{v}_2$ and the third generalized eigenvector $\boldsymbol{v}_3$ corresponding to the repeat eigenvalue $\lambda = 1$ span a subspace in which the dynamics of the return map do not simply decay back to

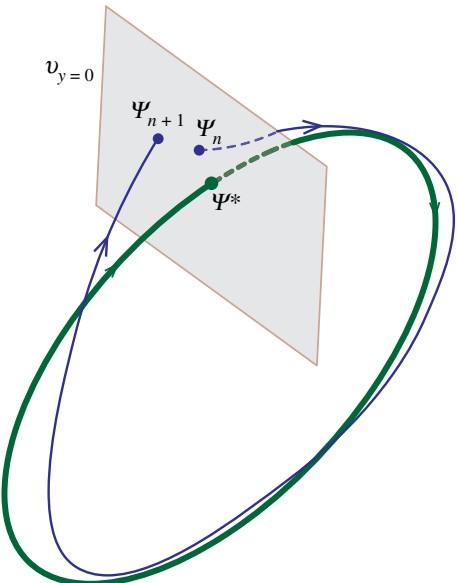

**Figure 5.** Illustration of the trajectory of the runner in state space in a reference frame that is translating along with the runner with velocity $v_{x0}$. The runner appears periodic in this reference frame and runner's mechanical state follows a periodic orbit. The return map $f_\bullet$ (● is 'ol' or 'an' for open-loop or anticipatory, respectively) is defined from the apex of the aerial phase ($v_y = 0$) to apex of the following aerial phase. $\boldsymbol{\psi}^*$ is the fixed point of the return map and $\boldsymbol{\psi}_n$ is a small perturbation away from the fixed point $\boldsymbol{\psi}^*$ at step $n$. In the next step, $\boldsymbol{\psi}_n$ maps to $\boldsymbol{\psi}_{n+1}$ at the apex of the following aerial phase under action of the return map $f_\bullet$.

the origin. For a diagonalizable system, any perturbation within this subspace would neither decay nor grow. However, the non-diagonalizable nature of $T_{an}$ leads to the outcome that a perturbation $\Delta\boldsymbol{\psi}_0$ within this subspace grows with increasing steps. The eigenvectors $\boldsymbol{v}_1$, $\boldsymbol{v}_2$, $\boldsymbol{v}_3$, the initial perturbation $\Delta\boldsymbol{\psi}_0$ and its growth after $n$ steps to $\Delta\boldsymbol{\psi}_n$ are given by

$$\boldsymbol{v}_1 = (0 \quad 0 \quad 1 \quad 0 \quad 0)^T, \quad \boldsymbol{v}_2 = (1 \quad 0 \quad 0 \quad 0 \quad 0)^T, \quad \text{and} \quad \boldsymbol{v}_3 = \left(0 \quad 0 \quad 0 \quad \tfrac{-1}{\sqrt{2}} \quad \tfrac{1}{\sqrt{2}}\right)^T, \quad (4.2a)$$

$$\Delta\boldsymbol{\psi}_0 = \sum_{k=1}^{3} \alpha_k v_k \tag{4.2b}$$

and $\quad \Delta\boldsymbol{\psi}_n = n\alpha_3\sqrt{2}\epsilon_n v_{y0}(\boldsymbol{v}_1 - \boldsymbol{v}_2) + \Delta\boldsymbol{\psi}_0, \quad$ respectively. $\tag{4.2c}$

As $n$ grows larger, the asymptotic approximation (denoted by $\approx$) is given by

$$\Delta\boldsymbol{\psi}_n \approx n\,\alpha_3\sqrt{2}\epsilon_n v_{y0} \begin{pmatrix} -1 \\ 0 \\ 1 \\ 0 \\ 0 \end{pmatrix}, \quad \text{where } n \gg 1. \tag{4.3}$$

Only a perturbation of magnitude $\alpha_3$ along $\boldsymbol{v}_3$ affects stability and leads to a nearly linear growth within the subspace spanned by the eigenvectors $\boldsymbol{v}_1$, $\boldsymbol{v}_2$. Perturbations along $\boldsymbol{v}_1$ or $\boldsymbol{v}_2$ neither grow nor decay because these represent invariance with respect to rotations and translations of the reference frame, respectively. A perturbation along the generalized eigenvector $\boldsymbol{v}_3$ may be geometrically viewed as one that conserves the velocity of the contact point on flat terrain but changes the angular momentum of the runner about its centre of mass. Therefore, any perturbation to the angular momentum will affect both orientation and forward speed.

For the special case of the anticipatory runner that completely avoids tangential collisions, the linearized return map $T_{an}$ with $\epsilon_{tc} = 1$ has eigenvalue $\lambda = 1$ of algebraic multiplicity 4 and geometric multiplicity 2, and one eigenvalue with $|\lambda| < 1$. The eigenvectors $\boldsymbol{v}_1$, $\boldsymbol{v}_2$ and the generalized

eigenvectors $\boldsymbol{v}_3$, $\boldsymbol{v}_4$ associated with $\lambda = 1$ form a basis for a subspace within which an initial perturbation $\Delta\boldsymbol{\psi}_0$ grows linearly with the number of steps $n$ in a subspace spanned by eigenvectors $\boldsymbol{v}_1$ and $\boldsymbol{v}_2$, i.e.

$$\boldsymbol{v}_1 = \begin{pmatrix} 0 \\ 0 \\ 1 \\ 0 \\ 0 \end{pmatrix}, \quad \boldsymbol{v}_2 = \begin{pmatrix} 1 \\ 0 \\ 0 \\ 0 \\ 0 \end{pmatrix}, \quad \boldsymbol{v}_3 = \begin{pmatrix} 0 \\ 0 \\ 0 \\ 0 \\ 1 \end{pmatrix}, \quad \text{and} \quad \boldsymbol{v}_4 = \begin{pmatrix} 0 \\ 0 \\ 0 \\ 1 \\ 0 \end{pmatrix}, \tag{4.4a}$$

$$\Delta\boldsymbol{\psi}_0 = \sum_{k=1}^{4} \alpha_k v_k, \quad \Delta\boldsymbol{\psi}_n = 2\epsilon_n v_{y0} n(\alpha_3 \boldsymbol{v_1} + \alpha_4 \boldsymbol{v_2}) + \Delta\boldsymbol{\psi}_0, \tag{4.4b}$$

$$\Delta\boldsymbol{\psi}_n \approx n2\epsilon_n v_{y0} \begin{pmatrix} \alpha_4 \\ 0 \\ \alpha_3 \\ 0 \\ 0 \end{pmatrix} \quad \text{for } n \gg 1. \tag{4.4c}$$

A perturbation to angular velocity $\omega$ causes a linear growth in orientation $\phi$, and a perturbation to the linear velocity $v_x$ causes a linear growth in position $x$. However, an anticipatory runner with $\epsilon_{tc} = 1$ avoids angular velocity perturbations due to the terrain altogether, i.e. $\alpha_3 = 0$. Therefore, only forward speed is affected due to the remaining unstable mode $\boldsymbol{v}_4$.

Although there are no unstable eigenvalues with magnitude greater than one, we find that the dynamics of running lead to an unstable growth with increasing steps. The growth due to non-diagonalizability of the return map is linearly proportional to the number of steps and not geometric as is the case for simple unstable eigenvalues. Importantly, the primary effect of the instability is to affect the forward speed and orientation, consistent with the numerical simulations that use finite perturbations and nonlinear dynamics. Also in agreement with the simulations, the only instability is translational when $\epsilon_{tc} = 1$.

# 5. Scaling analysis of the orientational failure mode

The mean steps to failure depends on many parameters, but none of the parameters separately predict the failure statistics (electronic supplementary material, figure S7). As most runners undergo orientational failures, we investigated whether the amount of body rotation accumulated over a single step due to a terrain slope perturbation would predict failure statistics.

If a runner with a periodic trajectory on flat ground encounters a sloped terrain of angle $\theta$, the orientation $\phi_\bullet$ at the next landing will no longer be vertical. This orientation $\phi_\bullet$ accumulated over one step depends on the take-off vertical velocity $v_{y,\bullet}^+$ via the aerial phase time $2v_{y,\bullet}^+$ and take-off angular velocity $\omega_\bullet^+$, as $\phi_\bullet = 2v_{y,\bullet}^+ \omega_\bullet^+$. The subscript '$\bullet$' is a placeholder for 'ol' or 'an' as the orientation change depends on whether the runner is purely open-loop (ol) or employs anticipatory (an) control. We hypothesize that the mean steps to failure $N_\bullet$ is a function of the orientational threshold $\phi_{tol}$ and the orientation change over a single step $\phi_\bullet$ alone, i.e. $N_\bullet = s_\bullet(\phi_{tol}, \phi_\bullet)$. Substituting the form of $s_\bullet(\phi_{tol}, \phi_\bullet)$ derived in electronic supplementary material, S8, we show that the mean steps to failure $N_\bullet$ is predicted to scale according to

$$N_\bullet \sim \frac{\phi_{tol}}{\phi_\bullet}, \tag{5.1}$$

where the expression for $\phi_\bullet$ is given in electronic supplementary material, equation (S15e).

The mean steps to failure in simulations performed with many different parameter values (electronic supplementary material, S7) are well-approximated by a single function of a dimensionless parameter $\phi_{tol}/\phi_\bullet$ (figure 6). The collapse of the simulation data highlights that the spin accumulated in one step due to a single perturbation (equation (5.1)) captures the fundamental principle underlying orientational failures. Importantly, this dimensionless parameter collapses the simulation data better than any individual parameter (electronic supplementary material, figure S7). Thus, the single parameter $\phi_{tol}/\phi_\bullet$ quantifies stability of runners of different sizes and mass distributions.

The dimensionless parameter $\phi_{tol}/\phi_\bullet$ also captures the parametric dependence of mean steps to failure on $\epsilon_{tc}$ and $\epsilon_n$ as seen from comparing the contour plots of $\phi_{tol}/\phi_\bullet$ shown in figure 7 against that of the direct simulations in figures 2d and 3a. The dependence of mean steps to failure on

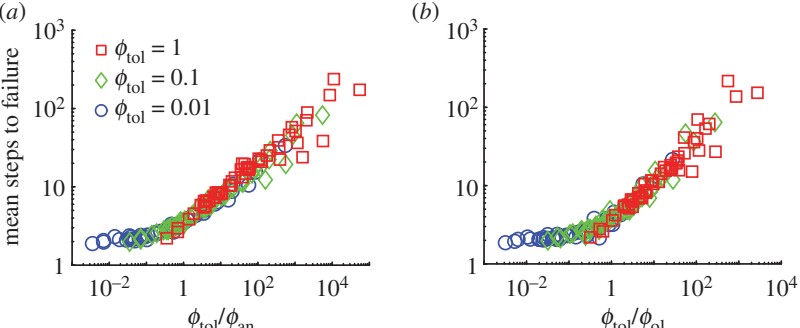

**Figure 6.** Generalizing results from §§3.1 and 3.4 to a wider range of physical and terrain parameters. Mean steps to failure from the Monte Carlo simulations is plotted against ($a$) $\phi_{tol}/\phi_{an}$ and ($b$) $\phi_{tol}/\phi_{ol}$ for different values of $\phi_{tol}$. The mean steps to failure depends mostly on a single dimensionless parameter $\phi_{tol}/\phi_{\bullet}$. All simulation parameters were varied independently in these simulations. But, for clarity, only variations in $\phi_{tol}$ are identified with different marker types.

$\epsilon_{tc}$ and $\epsilon_n$ for small slopes of the terrain is understood using a series expansion of $\phi_{\bullet}$ in terms of $\theta$ as given by

$$\phi_{ol} = \left(\frac{2v_{y0}^2}{1+I_{/G}}\right)\theta + \left(\frac{3-I_{/G}}{(1+I_{/G})^2} + \frac{4I_{/G}}{(1+I_{/G})^2}\epsilon_{tc} + \frac{2}{1+I_{/G}}\epsilon_n\right)v_{x0}v_{y0}\theta^2 + O(\theta^3) \tag{5.2a}$$

and

$$\phi_{an} = \left(\frac{2v_{y0}^2}{1+I_{/G}}(1-\epsilon_{tc})\right)\theta + \left(\frac{3-I_{/G}}{(1+I_{/G})^2} + \frac{5I_{/G}-3}{(1+I_{/G})^2}\epsilon_{tc} - \frac{4I_{/G}}{(1+I_{/G})^2}\epsilon_{tc}^2\right.$$
$$\left. + \frac{2(1-\epsilon_{tc})}{1+I_{/G}}\epsilon_n\right)v_{x0}v_{y0}\theta^2 + O(\theta^3). \tag{5.2b}$$

For the open-loop strategy, neither of the collision parameters, $\epsilon_n$ or $\epsilon_{tc}$, appear in the linear (leading order) term. When using an anticipatory strategy, the tangential collision parameter $\epsilon_{tc}$ appears to leading order. The normal collision parameter $\epsilon_n$ affects the second order dependence on $\theta$ for both strategies. These show why it is impossible to avoid orientational failures for the open-loop strategy but may be avoided when using the anticipatory strategy by choosing $\epsilon_{tc} = 1$.

For the open-loop runner, $\phi_{tol}/\phi_{ol}$ is independent of $\epsilon_n$ and $\epsilon_{tc}$ to first order in $\theta$ (equation (5.2a)). Hence, the contours in figure 7a (which resemble the contours in figure 2d from the Monte Carlo simulations) show a weak dependence on $\epsilon_n$ and $\epsilon_{tc}$ that arises from the $\theta^2$ term in equation (5.2a). The parameter $\phi_{ol}$ is smallest when $\epsilon_n = \epsilon_{tc} = 0$ and largest when $\epsilon_n = \epsilon_{tc} = 1$. For a human-like runner, $I_{/G} \ll 1$ (table 1), and thus the $\theta^2$ term in equation (5.2a) can be reduced to $(3 + 2\epsilon_n)v_{x0}v_{y0}\theta^2$, with no dependence on $\epsilon_{tc}$ at the asymptotic limit of $I_{/G} \ll 1$. The asymptotic analysis of $\phi_{ol}$ therefore explains why the contours of mean steps to failure in the Monte Carlo simulations are nearly parallel to the $\epsilon_{tc}$ axis and increase only slightly when $\epsilon_n$ is decreased (figure 2d).

For the anticipatory runner, the first-order term in the expansion depends on $\epsilon_{tc}$ (equation (5.2b)), unlike the case for the open-loop runner (equation (5.2a)). Nearly perfect anticipation corresponds to $\epsilon_{tc} \to 1$. At this limit $\phi_{an} \to 0$ and thus $N = \phi_{tol}/\phi_{an} \to \infty$, explaining why the contours of mean steps to failure in the Monte Carlo simulations are tightly bunched together in the neighbourhood of $\epsilon_{tc} = 1$ (figure 3a) and nearly parallel to the $\epsilon_n$ axis. Like for the open-loop runner, $\phi_{an}$ also shows a dependence on $\epsilon_n$ only in the pre-factor to the $\theta^2$ term of the series expansion in equation (5.2b). As $\epsilon_n$ decreases so does $\phi_{an}$, and thus $\phi_{tol}/\phi_{an}$ increases, capturing the trend observed in Monte Carlo simulations where decreasing $\epsilon_n$ increases steps taken for the anticipatory runner (figure 3a). For the anticipatory runner, unlike the open-loop runner, the $\epsilon_n$ dependence is coupled to $\epsilon_{tc}$, and thus the sensitivity of the parameter $\phi_{tol}/\phi_{\bullet}$ to changes in $\epsilon_n$ depend on the value of $\epsilon_{tc}$. The limit of $\epsilon_{tc} = 0$, where $\phi_{an} = \phi_{ol}$ has already been discussed above, for the open-loop runner. To analyse the case where $\epsilon_{tc} \to 1$, we approximate $\phi_{an}$ in the limit where $I_{/G} \ll 1$ (e.g. human-like runners) as

$$\phi_{an} \approx 2v_{y0}^2(1-\epsilon_{tc})\theta + (1-\epsilon_{tc})(2\epsilon_n + 3)v_{x0}v_{y0}\theta^2. \tag{5.3}$$

To understand the dependence of the mean steps to failure $N$ on $\epsilon_n$ and $\epsilon_{tc}$, we consider the limit of small angles of the terrain slope $\theta \ll 1$. Using equation (5.3), and for small $\theta$, we find that mean steps to

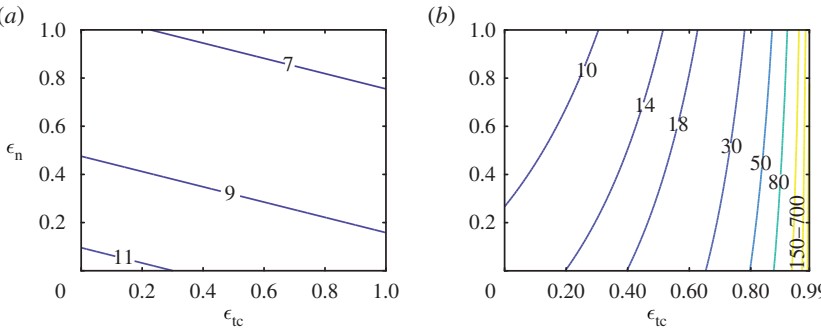

**Figure 7.** Contour plots of (a) $\phi_{\text{tol}}/\phi_{\text{ol}}$ and (b) $\phi_{\text{tol}}/\phi_{\text{an}}$ as a function of $\epsilon_n$ and $\epsilon_{tc}$ reveal that a single parameter captures the dependence of mean steps to failure of both open-loop runners (figure 2d) and anticipatory runners (figure 3a) on the collision parameters $\epsilon_n$ and $\epsilon_{tc}$. Recall that the only controllable parameters for the runners in these simulations are $\epsilon_n$ and $\epsilon_{tc}$. The complete expression for $\phi_\bullet$, shown in electronic supplementary material, equation (S15e), was used to generate these plots with parameter values drawn from table 1, and $\phi_{\text{tol}} = 1$. For the anticipatory runner, we restricted the maximum value of $\epsilon_{tc}$ to 0.99. For higher values of $\epsilon_{tc}$, orientational failures do not dominate and thus mean failure statistics are not accounted for by $\phi_{\text{tol}}/\phi_\bullet$.

failure $N = \phi_{\text{tol}}/\phi_{\text{an}}$ and its sensitivity to changes in $\epsilon_n$ are given by

$$N = \left(\frac{1}{1 - \epsilon_{tc}}\right) \frac{\phi_{\text{tol}}}{v_{y0}\theta}(2v_{y0} - 3v_{x0}\theta - 2v_{x0}\theta\epsilon_n) \tag{5.4a}$$

and

$$\frac{\partial N}{\partial \epsilon_n} = -\left(\frac{1}{1 - \epsilon_{tc}}\right) \frac{2\phi_{\text{tol}}v_{x0}}{v_{y0}}. \tag{5.4b}$$

Therefore, $N$ is more sensitive to changes in $\epsilon_n$ when $\epsilon_{tc} \to 1$. This resembles figure 3a where the mean steps to failure from the Monte Carlo simulations increases significantly as $\epsilon_n$ is reduced when $\epsilon_{tc} \to 1$, as opposed to when $\epsilon_{tc} \to 0$ where there is much lesser sensitivity of the mean steps to failure with respect to changes in $\epsilon_n$.

Improving running stability by increasing mean steps to failure helps provide more time for feedback-driven corrections in real-world runners. The analysis of mean steps to failure in the simplified runners without any feedback ability suggests that increasing $\phi_{\text{tol}}$ and decreasing $\phi_\bullet$ are both effective strategies to negotiate rough terrains. Therefore, besides altering $\epsilon_{tc}$ and $\epsilon_n$ in order to increase $\phi_{\text{tol}}/\phi_\bullet$ as already discussed, increasing $I_{/G}$ and reducing $v_{y0}$ also improves stability.

# 6. Discussion

We show that purely open-loop strategies with no feedback control cannot stabilize sagittal-plane dynamics during running. Such open-loop runners fail primarily by losing orientational stability and tumbling. Recall that for a chosen pair of $\epsilon_n$ and $\epsilon_{tc}$, the active push-off impulse is calculated for a flat terrain and re-used on every other terrain (§2.3). Using an anticipatory strategy to eliminate tangential collisions with the ground eliminates orientational instabilities but leads to a steady slowing down of the runner. However, on a step-like piecewise flat terrain, the open strategy is sufficient to stabilize the runner without losing forward speed, and so is the anticipatory strategy. If an anticipatory strategy is implemented noisily, i.e. the tangential collisions are low but not entirely eliminated, the runner suffers orientational instabilities. However, both the orientational and the translational instabilities are weak when using an anticipatory strategy and the growth of the instability is only linearly proportional to the number of steps taken and not a higher power. The exact number of steps to failure depend on many parameters including the inertial, geometry, collision parameters and the thresholds in orientation and speed for failure. This large set of parameters may be combined into a single dimensionless parameter that captures the failure statistics, which can also guide the morphological design of stable runners.

## 6.1. Impulsive stance assumption

An impulsive stance phase implies that the stance impulse is defined, but not the detailed time history of forces. Thus, the model may be used to study the dynamics and stability over multiple steps, but it cannot be used to find the actuation patterns that would achieve the desired impulse. Such simplified models may be used to specify the desired collisional and push-off impulses as constraints that should be met. More detailed models could then be used to calculate the stance force profiles as a constrained search or optimization problem.

The model also ignores the impulse due to the finite forces of gravity, because stance is treated as instantaneous. Relaxing the assumption of instantaneous stance implies that body weight affects the body's angular momentum about the contact point according to

$$H^+_{/P} - H^-_{/P} = \int_0^{T_{\text{stance}}} M_{/P}(t)\,dt, \tag{6.1}$$

where $T_{\text{stance}}$ is the stance duration and $M_{/P}(t)$ is the time-varying moment of the body weight about the contact point P. The torque due to gravitational forces about the contact point is proportional to body weight and the time-varying horizontal distance from the centre of mass to the contact point. The gravitational contribution is zero for a symmetric stance, and highest for the most asymmetric stance. Assuming a constant forward speed during stance and a $20°$ touchdown angle, the maximum change in angular momentum about the contact point, i.e. the integral in equation (6.1), is $|\Delta H_{/P}| \lessapprox 0.15$ in the same dimensionless units as before. For a typical human runner, the resultant orientation change in a single step is $\Delta\phi \lessapprox 0.01$, negligibly small compared to the influence of the terrain. Thus, ignoring torques induced by gravity has minimal impact on our conclusions.

A finite stance duration and the associated change in configuration allows a runner to control the body's sagittal-plane angular momentum independently from the forward and upward linear momenta. This may be understood in terms of breaking the symmetry of the stance phase or applying a large forward impulse and yet having the net ground reaction impulse pass through the centre of mass (no contribution to angular momentum). To not lose such control when considering an impulsive stance, we permit the application of an arbitrary angular impulse $J_\phi$ at the centre of mass during push-off (equation (2.1c)). Thus, having a finite stance duration and change of body configuration over stance is equivalent to $J_\phi$ (figure 8). The angular impulse also provides a means to accommodate torques due to a finite base of support and a moving centre of pressure during stance. However, recall that the constraint that the applied active push-off impulses should lead to perfectly periodic gait on flat terrain implies that $J_\phi = 0$. In our model, both open-loop and anticipatory runners slow down on rough terrain. Regaining forward speed needs a feedback controller and then the additional control authority offered by $J_\phi$ would be necessary to vary forward speed without affecting the body's angular momentum or vertical momentum.

## 6.2. Point contact assumption

Another limitation arises from considering a point-like contact that cannot capture effects associated with the spatial extent of the foot. These effects include the spatial filtering of terrain roughness and the application of a net torque about the initial contact point. The inclusion of $J_\phi$ in the model captures the application of torques, but there is no explicit means of incorporating the ability of the foot to act as a spatial filter [26]. Therefore, careful consideration should be given to the spatial frequency (wavenumber) of the roughness of the terrain when using a model with a point contact.

## 6.3. Timescale for feedback corrections

Open-loop runners with human-like parameters have a 99% chance of taking at least three steps without failing by exceeding the orientation threshold, while anticipatory runners ($\epsilon_{tc} = 1$) can take up to 15 steps with the same probability of completely losing forward momentum. This implies that the open-loop runner employing the slowest sensory modality (visual feedback delay $\approx 200$ ms [18]) has seven feedback cycles to correct for instabilities at endurance running speeds of $3\,\text{m s}^{-1}$ (step period $\approx 500$ ms [17]), with only an approximately 1% chance of an orientational failure. Thus, while sensory feedback is required to run on rough terrains, timescales associated with sensory feedback delays do not limit the runner's ability to maintain stability because of the slow-growing nature of the instability. Furthermore, employing an appropriate anticipatory strategy ($\epsilon_{tc} = 1$) eliminates the

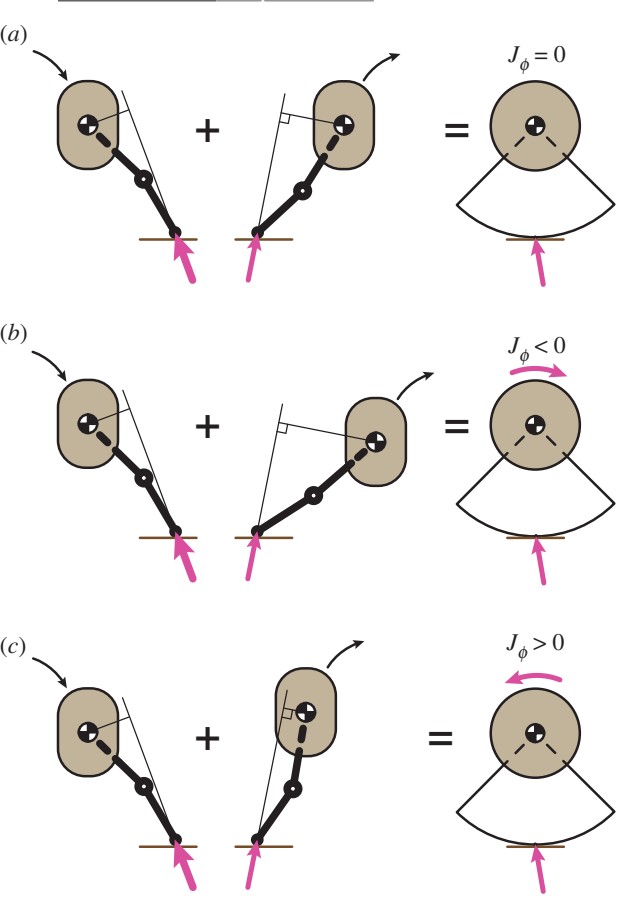

**Figure 8.** Equivalence between a runner using a finite stance with internal degrees of freedom versus an infinitesimal stance with an applied torque impulse. The contact of the general runner is represented as an impulse due to the collision at touchdown (left, pink vector) plus a net impulse due to the push-off (right, pink vector). In this example, by varying duration of stance, the runner can selectively vary the lever arm of the push-off impulse and thus the angular impulse about its centre of mass without altering the linear impulse. The net effect of a finite stance and change in configuration is therefore captured in our model (right column) by a linear impulse at the point of contact (pink vector at ground) and an angular impulse at the centre of mass ($J_\phi$).

orientational instability entirely, thereby further extending the timescale over which feedback is necessary.

## 6.4. Anticipation over one versus many steps

Our analyses are restricted to considerations of single-step anticipatory strategies. More general anticipatory strategies that rely on foreknowledge of the terrain for multiple steps have been previously considered for walking on rough terrains, both theoretically and experimentally [20,27]. In walking, Byl & Tedrake [20] found that a one-step lookahead policy performs almost as well as an optimal strategy that uses complete upcoming terrain information to plan footholds. Whether anticipatory strategies in running have similar diminishing gains with additional foreknowledge is beyond the scope of the present analysis. However, our formulation of the dynamics of running permits the inclusion of multi-step planning strategies by allowing the active impulses and the collision parameters to vary as a function of the oncoming terrain.

## 6.5. Leg retraction

Analyses of running models with leg mass suggest that optimal retraction rate is defined by stability demands, although these studies were limited to step-like terrains [7]. Experiments with runners on flat ground which measure the angle of the foot's velocity vector with respect to the ground [28]

suggest that foot velocity is perhaps not modulated in the manner we hypothesize. In the study by Blum *et al.* [28], the mean angle made by the subjects' foot velocity vector with the ground was 165°, whereas our prediction based on zero tangential speed of the foot at touchdown would imply that the angle should be 90°. Given these differences, we propose that the low values of $\hat{\epsilon}_t$ for human runners observed by Dhawale & Venkadesan [22] may result from joint stiffness modulation in the leg rather than precise control of the foot speed through leg retraction. Modulating foot and leg stiffness allows the runner to minimize the tangential collision and yet employ leg retraction strategies that accomplish other goals such as hypothesized by Seyfarth *et al.* [16] and Birn-Jeffery *et al.* [29].

## 6.6. Energy dissipation

Besides leg retraction, energy dissipation may also aid in stability based on studies of walking [30,31] and running [1,32]. Our model shows that while increasing energy dissipation in the direction normal to the terrain does increase the number of steps taken for open-loop and anticipatory runners, dissipating energy in the tangential collision is detrimental to stability. However, whether energy dissipation helps or hinders depends on the details of what is meant by 'open-loop'. For example, if the runner uses a *laboratory-fixed* push-off policy instead of the *terrain-fixed* push-off described in the main text, dissipating energy in the normal direction is also detrimental to orientational stability (electronic supplementary material, S10). Thus, the hypothesized trade-off between energy consumption and stability is not universally true in our models. Our results are consistent with experiments on running birds encountering sudden terrain drops, as the birds do not always dissipate energy on the perturbation step [1]. Our results might provide a means to understand why the increase in energy consumption for humans running on step-like terrains is only 5% [4]. We find that open-loop strategies are sufficient to maintain stability on step-like terrains and additional energy expenditure provides little added benefit.

## 6.7. Implications of scaling analysis to body plan of animals

The single parameter $\phi_{tol}/\phi_\bullet$ that predicts mean steps to failure (figure 6) generalizes our results beyond runners with human-like parameters and morphology, and thus can be used as a criteria to assess a runner's stability. This is because runners with a larger $\phi_{tol}/\phi_\bullet$ should be able to maintain orientation for a greater number of steps in the absence of sensory feedback control.

The parameter $\phi_{tol}$ is the maximum angle of tilt the runner can accumulate before it falls. Animals that possess larger (base/height) ratios, i.e. have a landscape rather than portrait orientation when viewed in the sagittal plane, increase $\phi_{tol}$ and thereby increase the parameter $\phi_{tol}/\phi_\bullet$. Quadrupeds such as cats and dogs and other adept runners such as cockroaches possess such an aspect ratio. Larger range of motion of the leg with respect to the body also increases $\phi_{tol}$ as it improves the ability of the animal to correct for body tilt by placing the foot in front of the runner when initiating stance. In our simulations, the choice of $\phi_{tol}$ value is based on this consideration of the leg angle for humans. Other adept runners with portrait orientations such as ostriches also possess a large range of motion of their legs. Penguins, which are not known to be adept runners, occupy the lower end of the $\phi_{tol}$ scale due to possessing a portrait orientation as well as low range of motion of their legs compared to other bipeds such as humans, turkeys and ostriches.

Low take-off angles, for a given forward speed, and increased radius of gyration $r_g$ relative to leg length $r_\ell$ would be beneficial to stability as $\phi_{tol}/\phi_\bullet \propto I_{/G}/v_{y0}^2$ (equations (5.2)). However, very low take-off angles increase the risk of tripping on rough terrains. A larger radius of gyration could be achieved by increasing distal masses in appendages like arms and legs. However, increasing distal masses in the leg increases the metabolic cost of running [33] via increased energetic cost associated with swinging the leg [34,35] and may also lead to potentially injurious collisions. Running animals like horses and bison have light legs, but increase $r_g$ by possessing extended torsos and large heads. Tails in animals like cats and lizards, which are used in active stabilization [36], are yet another means to increase $r_g$, thereby reducing $\phi_\bullet$. Anticipatory runners further benefit from setting $\epsilon_t \approx 1$ like observed in humans [22], thereby drastically reducing $\phi_\bullet$ as discussed in §5. Thus, the dimensionless parameter $\phi_{tol}/\phi_\bullet$ is qualitatively consistent with the body morphology of adept and poor animal runners, and we propose that it can be used as a design criteria for running robots.

# 7. Notation

Scalars are denoted by italic symbols (e.g. $m$ for mass of the runner and $I$ for the moment of inertia), vectors by bold, italic symbols ($\boldsymbol{J}_{\mathrm{imp}}$ for push-off impulse and $\boldsymbol{v}$ for velocity), points or landmarks in capitalized non-italic symbols (such as centre of mass G in figure 1) and capitalized, bold, non-italic symbols for matrices (such as return map matrix $\boldsymbol{T}_{\mathrm{an}}$). Vectors associated with a point, such as velocity of centre of mass G are written as $\boldsymbol{v}_{\mathrm{G}}$, with the uppercase alphabet in the subscript specifying the point in the plane. Angular momentum vectors or moment of inertia variables are subscripted with '/A', representing angular momentum or moment of inertia computed about point A, such as $I_{/\mathrm{G}}$ representing moment of inertia about centre of mass G. The $\hat{x}$ component of the velocity vector of point A, i.e. $\vec{v}_{\mathrm{A}}$ is denoted with a subscript 'A,x', e.g. tangential velocity of the contact point P is written as $v_{\mathrm{P,t}}$. The symbols $v_{x0}$, $v_{y0}$ denote the initial forward and vertical velocities of the runner at take-off. Variables just before collision with the terrain are denoted by the superscript '$-$', after passive collision but before push-off by the superscript 'c' and just after push-off by the superscript '+'. For example, angular velocity before collision is $\omega^{-}$, after passive collision is $\omega^{c}$ and just after push-off is $\omega^{+}$.

Data accessibility. All detailed derivations are included in the electronic supplementary material. No other data were generated in this research.

Authors' contributions. M.V. conceived of the model. N.D. and M.V. ran the simulations. All authors were involved in the analysis of the model; S.M., M.V. and N.D. performed the linear stability analysis, M.V. and N.D. performed the one-step analysis to capture mean statistics, and M.V., S.M. and N.D. did the analysis of the steps to failure distributions. N.D. and M.V. wrote the manuscript, and all authors edited it.

Competing interests. We declare we have no competing interests.

Funding. This work was funded by the Human Frontier Science Program and the Wellcome/DBT India Alliance.

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
