## [Reviewer comments · Royal Society Open Science]

Review History

RSOS-181729.R0 (Original submission)

Review form: Reviewer 1

Is the manuscript scientifically sound in its present form?

Yes

Are the interpretations and conclusions justified by the results?

Yes

Is the language acceptable?

Yes

Is it clear how to access all supporting data?

No

Do you have any ethical concerns with this paper?

No

Have you any concerns about statistical analyses in this paper?

No

Recommendation?

Accept with minor revision (please list in comments)

Comments to the Author(s)

This manuscript is about the development, simulation, and analysis of a running model to understand the stability of running on uneven ground. The disc-like model was tested with both an open-loop strategy (without sensory feedback) and an anticipatory strategy that serves to minimize tangential collisions. Instabilities related to body orientation (orientational instability) and forward motion (translational instability) were explored with Monte Carlo simulations. The authors found that open-loop strategies cannot handle rough terrain due to orientational stability but can manage step-like terrain. In contrast, anticipatory strategies can achieve orientational stability and handle rough terrain but lose translational stability through loss of forward speed. Further analysis showed that mean steps to failure can be captured by the ratio between the orientational threshold and the orientation change over one step.

This manuscript was generally well-written and clear. The graphs are descriptive and help support the points mentioned in the text. The authors were thorough in explaining their method, results, and providing further linear stability analysis to understand their results. The flow of derivations and explanations was generally smooth despite 28 references to the supplementary material. The authors also included a constructive discussion on the merits and limitations of their model and their results in the context of biological behavior. I only have minor comments.

Minor Comments

Title is short and pithy but a bit too general. Perhaps it would benefit from a key takeaway from the results.

It would serve the reader better to mention earlier in the manuscript that the model was disc-like, closer to the beginning of Section 2 than the end. I puzzled over Figure 1b for a while trying to figure out if the disc was intentional or not until I reached the last few sentences of Section 2.

Figure 1a would be clearer with an actual aerial phase shown in the human outline.

A few items are a bit unclear. It is probably worth repeating in the beginning of Section 3 that first the impulses from flat ground on determined for steady-state running and then maintained for the rough terrain simulations. The parameters being adjusted in the simulation are not clear. Perhaps repeating the e_{tc} was varied would be useful. It is also unclear how angular impulse was maintained. The angular impulse was mentioned early in the manuscript but not again until near the end. Was it always set to 0?

Page 9 Line 208/47: It is unclear which experiments "in experiments" is being referred. Is this referring to the simulation? Or referencing human or animal experiments?

Page 16 Line 322/16: do not, instead of "don't"

Review form: Reviewer 2

Is the manuscript scientifically sound in its present form?

Yes

Are the interpretations and conclusions justified by the results?

Yes

Is the language acceptable?

Yes

Is it clear how to access all supporting data?

No

Do you have any ethical concerns with this paper?

No

Have you any concerns about statistical analyses in this paper?

No

Recommendation?

Accept with minor revision (please list in comments)

Comments to the Author(s)

The paper addresses open-loop and anticipatory strategies for running on rough terrain using a simple math model. They perform Monte Carlo simulations to assess the likelihood of failure using open-loop and “anticipatory” strategies. The authors correlate successful running to practical dynamical concerns, like the degree of energy dissipation per step. I believe it is a well-thought-out contribution and would only require minor changes in framing to merit publication. The material is thoroughly communicated and well-written. The supplementary materials are similarly thorough. The locomotion math model is well motivated and elegantly designed to capture some of the key dynamics of running. I am heartened the authors are using “mean steps to failure” as a stability metric, because it seems most appropriate in the context of stochastic terrain. The method of modeling noises in anticipatory running is also solid.

I would like to see the authors address the limits of their “anticipatory strategy.” The authors rightly reference Byl and Tedrake for using a “look-ahead” strategy. I’m not convinced that the authors’ anticipatory strategy is sufficiently powerful of a control model to reflect an optimal running behavior, or what I would consider truly “anticipatory” over multiple steps. I would like to see the authors draw a direct contrast between their anticipatory approach and the Byl 2009 approach when describing their control modeling approach.

The title is succinct and punchy, but perhaps overly broad. This paper primarily addresses relatively open-loop strategies to running on rough terrain. There are, by contrast, approaches that use high-level planning that could plan over many steps of rough terrain. Also, this glosses over a lot of the control complexity associated with runners with multi-body models, specifically those with compliance. I would recommend including a reference in the title to the fact that you’re analyzing simple models, maybe “How to run on rough terrains: lessons from a simple model.” Obviously, the style of title would be up to the authors, but I think that the study does need to be bounded in scope a bit given the complexity of the field.

Decision letter (RSOS-181729.R0)

23-Jan-2019

Dear Professor Venkadesan

On behalf of the Editors, I am pleased to inform you that your Manuscript RSOS-181729 entitled "How to run on rough terrains" has been accepted for publication in Royal Society Open Science subject to minor revision in accordance with the referee suggestions. Please find the referees' comments at the end of this email.

The reviewers and handling editors have recommended publication, but also suggest some minor revisions to your manuscript. Therefore, I invite you to respond to the comments and revise your manuscript.

- Ethics statement

- Data accessibility

<http://datadryad.org/submit?journalID=RSOS&manu=RSOS-181729>

- Competing interests

- Authors' contributions

AB carried out the molecular lab work, participated in data analysis, carried out sequence alignments, participated in the design of the study and drafted the manuscript; CD carried out

the statistical analyses; EF collected field data; GH conceived of the study, designed the study, coordinated the study and helped draft the manuscript. All authors gave final approval for publication.

- Acknowledgements

- Funding statement

Because the schedule for publication is very tight, it is a condition of publication that you submit the revised version of your manuscript before 01-Feb-2019. Please note that the revision deadline will expire at 00.00am on this date. If you do not think you will be able to meet this date please let me know immediately.

- 1) A text file of the manuscript (tex, txt, rtf, docx or doc), references, tables (including captions) and figure captions. Do not upload a PDF as your "Main Document";
- 2) A separate electronic file of each figure (EPS or print-quality PDF preferred (either format should be produced directly from original creation package), or original software format);
- 3) Included a 100 word media summary of your paper when requested at submission. Please ensure you have entered correct contact details (email, institution and telephone) in your user account;
- 4) Included the raw data to support the claims made in your paper. You can either include your data as electronic supplementary material or upload to a repository and include the relevant doi within your manuscript. Make sure it is clear in your data accessibility statement how the data can be accessed;

5) All supplementary materials accompanying an accepted article will be treated as in their final form. Note that the Royal Society will neither edit nor typeset supplementary material and it will be hosted as provided. Please ensure that the supplementary material includes the paper details where possible (authors, article title, journal name).

on behalf of Dr Monica Daley (Associate Editor) and Professor Kevin Padian (Subject Editor)
openscience@royalsociety.org

Associate Editor Comments to Author (Dr Monica Daley):

Thank you for your patience in waiting for a decision on your paper. We have now received feedback from two expert reviewers, who are positive about the contribution of this paper in developing a simple model of running on uneven ground. In particular they highlight that the paper makes a novel contribution through the combination of simple models and stochastic approaches. The reviewers have suggested only minor changes to the text; therefore I am happy to accept the paper subject to these minor revisions. Please provide a point-by-point response to the reviewers when resubmitting the revised version.

Reviewer comments to Author:
Reviewer: 1

Comments to the Author(s)

This manuscript is about the development, simulation, and analysis of a running model to understand the stability of running on uneven ground. The disc-like model was tested with both

an open-loop strategy (without sensory feedback) and an anticipatory strategy that serves to minimize tangential collisions. Instabilities related to body orientation (orientational instability) and forward motion (translational instability) were explored with Monte Carlo simulations. The authors found that open-loop strategies cannot handle rough terrain due to orientational stability but can manage step-like terrain. In contrast, anticipatory strategies can achieve orientational stability and handle rough terrain but lose translational stability through loss of forward speed. Further analysis showed that mean steps to failure can be captured by the ratio between the orientational threshold and the orientation change over one step.

This manuscript was generally well-written and clear. The graphs are descriptive and help support the points mentioned in the text. The authors were thorough in explaining their method, results, and providing further linear stability analysis to understand their results. The flow of derivations and explanations was generally smooth despite 28 references to the supplementary material. The authors also included a constructive discussion on the merits and limitations of their model and their results in the context of biological behavior. I only have minor comments.

Minor Comments

Title is short and pithy but a bit too general. Perhaps it would benefit from a key takeaway from the results.

It would serve the reader better to mention earlier in the manuscript that the model was disc-like, closer to the beginning of Section 2 than the end. I puzzled over Figure 1b for a while trying to figure out if the disc was intentional or not until I reached the last few sentences of Section 2.

Figure 1a would be clearer with an actual aerial phase shown in the human outline.

A few items are a bit unclear. It is probably worth repeating in the beginning of Section 3 that first the impulses from flat ground on determined for steady-state running and then maintained for the rough terrain simulations. The parameters being adjusted in the simulation are not clear. Perhaps repeating the e_{tc} was varied would be useful. It is also unclear how angular impulse was maintained. The angular impulse was mentioned early in the manuscript but not again until near the end. Was it always set to 0?

Page 9 Line 208/47: It is unclear which experiments "in experiments" is being referred. Is this referring to the simulation? Or referencing human or animal experiments?

Page 16 Line 322/16: do not, instead of "don't"

Reviewer: 2

Comments to the Author(s)

The paper addresses open-loop and anticipatory strategies for running on rough terrain using a simple math model. They perform Monte Carlo simulations to assess the likelihood of failure using open-loop and "anticipatory" strategies. The authors correlate successful running to practical dynamical concerns, like the degree of energy dissipation per step. I believe it is a well-thought-out contribution and would only require minor changes in framing to merit publication. The material is thoroughly communicated and well-written. The supplementary materials are similarly thorough. The locomotion math model is well motivated and elegantly designed to capture some of the key dynamics of running. I am heartened the authors are using "mean steps to failure" as a stability metric, because it seems most appropriate in the context of stochastic terrain. The method of modeling noises in anticipatory running is also solid.

I would like to see the authors address the limits of their “anticipatory strategy.” The authors rightly reference Byl and Tedrake for using a “look-ahead” strategy. I’m not convinced that the authors’ anticipatory strategy is sufficiently powerful of a control model to reflect an optimal running behavior, or what I would consider truly “anticipatory” over multiple steps. I would like to see the authors draw a direct contrast between their anticipatory approach and the Byl 2009 approach when describing their control modeling approach.

The title is succinct and punchy, but perhaps overly broad. This paper primarily addresses relatively open-loop strategies to running on rough terrain. There are, by contrast, approaches that use high-level planning that could plan over many steps of rough terrain. Also, this glosses over a lot of the control complexity associated with runners with multi-body models, specifically those with compliance. I would recommend including a reference in the title to the fact that you’re analyzing simple models, maybe “How to run on rough terrains: lessons from a simple model.” Obviously, the style of title would be up to the authors, but I think that the study does need to be bounded in scope a bit given the complexity of the field.

Author's Response to Decision Letter for (RSOS-181729.R0)

See Appendix A.

Decision letter (RSOS-181729.R1)

15-Feb-2019

Dear Professor Venkadesan,

I am pleased to inform you that your manuscript entitled "Dynamics and stability of running on rough terrains" is now accepted for publication in Royal Society Open Science.

on behalf of Dr Monica Daley (Associate Editor) and Kevin Padian (Subject Editor)
openscience@royalsociety.org

Associate Editor Comments to Author (Dr Monica Daley):

Thank you for thoroughly addressing the comments raised by the expert reviewers. The revised text is thorough and clear.

Appendix A

Dear editors and reviewers,

We are delighted that our manuscript has been provisionally accepted, and thank the reviewers for their careful comments. Please find here our responses to these comments and the revised version of the manuscript.

Associate Editor comments to Authors:

=====

"Thank you for your patience in waiting for a decision on your paper. We have now received feedback from two expert reviewers, who are positive about the contribution of this paper in developing a simple model of running on uneven ground. In particular they highlight that the paper makes a novel contribution through the combination of simple models and stochastic approaches. The reviewers have suggested only minor changes to the text; therefore I am happy to accept the paper subject to these minor revisions. Please provide a point-by-point response to the reviewers when resubmitting the revised version."

We appreciate the effort taken by the reviewers in helping to improve the manuscript. Presented below is a point-wise response to the requested revisions. Both reviewers recommended that the title be modified. So, we respond to that first, followed by point-wise responses to each reviewer.

Reviewer comments to Author:

=====

Title:

* Reviewer 1: "Title is short and pithy but a bit too general. Perhaps it would benefit from a key takeaway from the results."

* Reviewer 2: "The title is succinct and punchy, but perhaps overly broad. This paper primarily addresses relatively open-loop strategies to running on rough terrain. There are, by contrast, approaches that use high-level planning that could plan over many steps of rough terrain. Also, this glosses over a lot of the control complexity associated with runners with multi-body models, specifically those with compliance. I would recommend including a reference in the title to the fact that you're analyzing simple models, maybe "How to run on rough terrains: lessons from a simple model." Obviously, the style of title would be up to the authors, but I think that the study does need to be bounded in scope a bit given the complexity of the field."

We appreciate the need to balance brevity with specificity. With that in mind, the new title is, "Dynamics and stability of running on rough terrains." This title reflects the features of rough terrain running that we analyze in this manuscript.

Our point-wise responses to the reviewers follow.

Reviewer 1:

1. "It would serve the reader better to mention earlier in the manuscript that the model was disc-like, closer to the beginning of Section 2 than the end. I puzzled over Figure 1b for a while trying to figure out if the

disc was intentional or not until I reached the last few sentences of Section 2"

Line 88 changed to - "We model the runner in the sagittal plane as a disc-like rigid body..."

2. "Figure 1a would be clearer with an actual aerial phase shown in the human outline."

The illustration of a human runner is based on motion capture data from a real human runner. We included it in its present form so that the reader may visually assess the change in configuration of the body during stance. In light of this reviewer's comment, we now choose the third time-point to precede touchdown so that the runner is aerial. We have also modified the center of mass trajectory to use different color and stroke types for the stance and aerial phases.

3. "It is probably worth repeating in the beginning of Section 3 that first the impulses from flat ground on determined for steady-state running and then maintained for the rough terrain simulations."

The paragraph that immediately precedes section 3 states this in lines 148-150 and 156-169. However, we can see how a well-placed repetition of this detail will aid the reader. Therefore, we remind the reader in the discussion section on line 432-433: "Recall that for a chosen pair of ϵ_n and ϵ_{tc} , the active push-off impulse is calculated for a flat terrain and reused on every other terrain (section 2.3)."

4. "The parameters being adjusted in the simulation are not clear. Perhaps repeating the ϵ_{tc} was varied would be useful."

The start of Section 3, (lines 178-179, Monte Carlo results) has been modified to state, "We perform Monte Carlo simulations on step-like terrains and undulating rough terrains to estimate the statistics of failure for both open-loop and anticipatory runners. These simulations use different combinations of the normal and tangential collision parameters, ϵ_n and ϵ_{tc} , respectively."

5. "It is also unclear how angular impulse was maintained. The angular impulse was mentioned early in the manuscript but not again until near the end. Was it always set to 0?"

The angular impulse is always set to 0 because the active linear and angular impulses were found for a steady, periodic gait on flat terrain and thereafter held fixed. We append these sentences to the end of section 2: "The active linear and angular impulses were found for a steady, periodic gait on flat terrain and held fixed thereafter. For a steady gait, the angular impulse $J_\phi = 0$ at all times, whereas the linear impulse depends on the collision parameters and speed. Section 6 examines the role of J_ϕ in extending the model to account for finite stance durations."

6. "Page 9 Line 208/47: It is unclear which experiments "in experiments" is being referred. Is this referring to the simulation? Or referencing human or animal experiments?"

This sentence, now on line 207 has been rewritten as, "Future experiments that track foot placement on rough terrains could test whether a runner uses a strategy of aiming for flat regions of the terrain by estimating the fraction of footfalls on flat versus highly sloped regions."

7. "Page 16 Line 322/16: do not, instead of "don't""

Done.

Reviewer 2:

1. "I would like to see the authors address the limits of their "anticipatory strategy." The authors rightly reference Byl and Tedrake for using a "look-ahead" strategy. I'm not convinced that the authors' anticipatory strategy is sufficiently powerful of a control model to reflect an optimal running behavior, or what I would consider truly "anticipatory" over multiple steps. I would like to see the authors draw a direct contrast between their anticipatory approach and the Byl 2009 approach when describing their control modeling approach"

Thank you for the suggestion. We have added a new paragraph, 'Anticipation over one versus many steps', from line 498, following the discussion subsection 'Timescale for feedback corrections' to frame our model relative to that of Byl and Tedrake, 2009.

[revised manuscript text omitted]

$H_{/P}^c - H_P^- = 0,$ (1b)

push-off: $\mathbf{v}_G^+ = \mathbf{v}_G^c + \mathbf{J}_{\text{imp}}, \quad I_G \omega^+ = I_G \omega^c + J_{\text{imp},t} + J_\phi,$ (1c)

flight: $\ddot{x}_G(t) = 0, \quad \ddot{y}_G(t) = -1, \quad \ddot{\phi}(t) = 0,$ and (1d)

initial conditions: $\begin{pmatrix} x_G(0) \\ y_G(0) \\ \phi(0) \end{pmatrix} = \begin{pmatrix} x_G^+ \\ y_G^+ \\ \phi^+ \end{pmatrix}, \quad \begin{pmatrix} \dot{x}_G(0) \\ \dot{y}_G(0) \\ \dot{\phi}(0) \end{pmatrix} = \begin{pmatrix} v_{G,x}^+ \\ v_{G,y}^+ \\ \omega^+ \end{pmatrix}.$ (1e)

Horizontal and vertical positions are denoted by x and y , respectively, orientation by ϕ , velocity
by \mathbf{v} , angular velocity by ω , and angular momentum by H . Subscripts of ‘t’ and ‘n’ refer to the
tangential and normal directions to the terrain, while subscripts of ‘x’ and ‘y’ refer to the horizontal
and vertical directions in the lab frame (Fig. 1b). Superscript ‘-’ denotes variables immediately
preceding the collision, ‘c’ after the passive collision and ‘+’ after the active push-off. Subscripts
P and G refer to quantities associated with the contact point and center of mass, respectively.

The mechanical state of the runner is parameterized by the center of mass positions (x_G, y_G) ,
body orientation ϕ , and their respective velocities $(v_{G,x}, v_{G,y})$ and ω . Because the stance is assumed
to be instantaneous, the velocities may change discontinuously but the position and orientation
remain constant during stance. The instantaneous stance assumption also implies that finite forces
such as gravity or air-drag do not contribute to the impulse on the runner. However, the active
rotational impulse J_ϕ applied at the center of mass G captures the effects of varying posture over
stance and the changing center of pressure on the ground. We examine this approximation and its
implications in the discussion.

**2.2 Stance: passive collision**

The passive collision has two components, the normal and the tangential, which are parametrized
by ϵ_n and ϵ_t , respectively. Modulating these coefficients of restitution would exert control over the
passive collision. We hold these parameters constant from step-to-step and examine the effect of
different values on stability.

An animal may vary the extent to which the normal momentum impinging on the terrain is
stored elastically and later used for push-off. Without detailed consideration of how an animal may
accomplish this, we allow our model to precisely specify ϵ_n . The normal collision is perfectly elastic
for $\epsilon_n = 1$ and perfectly inelastic for $\epsilon_n = 0$.

The tangential collision directly affects the body’s angular momentum. We therefore elaborate
the model to consider different ways in which this collision may be controlled. One strategy for an
animal to modulate the tangential collision is to vary the foot speed along the terrain’s tangential
direction. For the same speed of the center of mass, the foot speed may be varied through actions
such as retracting the leg. However, control of the foot’s speed relative to the terrain’s tangential
direction requires sensory information on the ground speed and the terrain’s slope. In our model,

this corresponds to knowledge of $v_{P,t}^-$, the speed of the contact point along the terrain’s tangent
 in the moment just before the collision. We consider two extremes for the availability of such
 sensory information. At one extreme is an *open-loop strategy* that has no information and assumes
 $v_{P,t}^- = v_{x0}$, which implies three assumptions: forward speed equals the initial speed, the terrain is
 flat, and the body has no spin. In this case, the intended tangential coefficient of restitution ϵ_{tc}
 and the actual coefficient ϵ_t would generally not equal each other and depend on the ratio $v_{x0}/v_{P,t}^-$.
 At the other extreme is an *anticipatory strategy* with perfect *a priori* information so that $\epsilon_{tc} = \epsilon_t$.
 These are summarized as,

$$141 \quad \epsilon_t = \begin{cases} \epsilon_{tc} \frac{v_{x0}}{v_{P,t}^-} & : \text{open-loop,} \\ \epsilon_{tc} & : \text{anticipatory.} \end{cases} \quad (2)$$

We derive this equation (2) more formally in supplement S3.2, and examine the relationship between
 ϵ_{tc} and ϵ_t for the open-loop strategy in supplement S4.1.

**2.3 Stance: active push-off**

Stance ends with the application of an active, linear push-off impulse \mathbf{J}_{imp} at the contact point P
 and an active angular push-off impulse J_ϕ at the center of mass G. We constrain these impulses so
 that in the absence of external perturbations or other disturbances the runner is perfectly periodic
 and remains upright ($\phi(t) = 0$) on flat ground. Importantly, once the impulses are chosen for flat
 ground, they are not allowed to vary step-to-step on any other terrain to reflect the absence of
 active feedback control. Together, these conditions imply that the active push-off impulses \mathbf{J}_{imp}
 and J_ϕ depend only on ϵ_n , ϵ_{tc} , v_{x0} and v_{y0} , and no other parameters, according to

$$152 \quad \mathbf{J}_{\text{imp}} = \begin{pmatrix} v_{x0} \\ v_{y0} \end{pmatrix} - \begin{pmatrix} (\epsilon_{tc} + \frac{1-\epsilon_{tc}}{1+I/G})v_{x0} \\ -\epsilon_n v_{y0} \end{pmatrix}, \quad (3a)$$

$$153 \quad J_\phi = 0. \quad (3b)$$

Thus the center of mass of a periodic runner on flat ground has a constant forward speed v_{x0} and
 vertical speed v_{y0} at every step.

On rough terrains, there are two options for defining the application of the invariant linear
 impulse on every step. First, the impulse vector \mathbf{J}_{imp} may be held constant in every step with

respect to gravity ($\hat{x} - \hat{y}$ frame in Fig. 1b), which we call the *lab-fixed* push-off policy. Second,
 the impulse vector may be held constant in every step with respect to the normal direction to the
 terrain at the point of contact ($\hat{t} - \hat{n}$ frame in Fig. 1b), which we call a *terrain-fixed* push-off policy.
 The terrain-fixed policy may be considered a better approximation of what animals do, because
 the normal to the terrain and the leg orientation are often coupled, whereas leg orientation at
 contact and gravity may vary from step to step. Implicit in preferring the terrain-fixed policy is the
 assumption that joint torques to apply forces are planned in an ego-centric (body-fixed) frame of
 reference. For the disc-like model of a runner that we use, the terrain-fixed and body-fixed policies
 are identical. Detailed expressions for the velocities in the stance phase, as well as expressions for
 \mathbf{J}_{imp} under both push-off policies are given in supplement S3. We present a complete analysis of
 the lab-fixed push-off policy in supplement S10 and focus on the terrain-fixed policy in the main
 paper. The active linear and angular impulses were found for a steady, periodic gait on flat terrain
 and held fixed thereafter. For a steady gait, the angular impulse \$J_\phi = 0\$ at all times, whereas the
 linear impulse depends on the collision parameters and speed. Section 6 examines the role of \$J_\phi\$ in
 extending the model to account for finite stance durations.

3 Monte Carlo simulations

A sagittal plane runner can only fail by two modes, when the body orientation exceeds a chosen
 threshold (*orientational failure*), or by failing to move forward any longer (*translational failure*).
 We choose the orientational threshold ϕ_{tol} as the angle of tilt to passively topple a human who is
 standing with their feet apart in a pose resembling double-stance in walking.

We perform Monte Carlo simulations on step-like terrains and undulating rough terrains to
 estimate the statistics of failure for both open-loop and anticipatory runners. These simulations
 use different combinations of the normal and tangential collision parameters, \$\epsilon_n\$ and \$\epsilon_{tc}\$, respectively.
 Stability is quantified by the mean steps to failure, like previous studies of rough terrain walking
 [20].

The terrain is modeled as a piecewise linear interpolation of an underlying random grid. The grid
 points are separated by a distance λ and the heights h of the grid points are chosen from a uniform
 random distribution (table 1, supplement S1.3). A linear interpolation between the grid points
 yields a terrain with random variations in both slope and height. Corners at grid point implies an

Runner					Terrain	Monte Carlo		
I/G	ϵ_n	ϕ_{tol}	v_{x0}	v_{y0}	h	λ	M	MAX
0.17	0.63	$\pi/6$	0.96	0.26	$\sim \mathcal{U}(-0.03, 0.03)$	0.1	10^5	10^3

[revised manuscript text omitted]

$$256 \quad \epsilon_{t,\text{noisy}} = \epsilon_{tc} + \Delta\epsilon_t\eta, \quad (4a)$$

$$257 \quad \text{where } \eta \sim \mathcal{U}[-1, 1], \Delta\epsilon_t \in \mathbb{R}. \quad (4b)$$

The uniformly distributed zero-mean random variable η models random step-to-step noise in ϵ_{tc}
 and $\Delta\epsilon_t$ is the noise intensity. The noisy $\epsilon_{t,\text{noisy}}$ is allowed to exceed 1 in our simulations.

We find that incurring tangential collisions ($\epsilon_{tc} < 1$) is optimal when there is non-zero noise
 ($\Delta\epsilon_t > 0$). This is unlike the noiseless anticipatory runner whose optimum is $\epsilon_{tc} = 1$. Moreover,
 noise in controlling tangential collisions does affect stability and the mean steps to failure are
 severely reduced (Fig. 3b). For example, compared to a noiseless human-like runner, the mean
 steps to failure drops nine-fold for a runner with noise intensity $\Delta\epsilon_t = 0.1$, and the optimum ϵ_{tc}
 decreases by 1% to $\epsilon_{tc} = 0.99$ (Fig. 3b). Additional noise in the tangential collision of open-loop
 runners reduces the number of steps taken, but does not alter the dependence of steps taken on ϵ_{tc}
 (supplement S4.2). Therefore, for anticipatory runners, noise in controlling the tangential collision
 implies that incurring a slight tangential collision is optimal but at the cost of stability.

**3.6 Predictions for ϵ_t in experiments**

A main finding of our analyses is the importance of minimizing tangential collisions with the ground
 when running on rough terrains. But measuring ϵ_t on rough terrains is challenging because it needs
 a well-defined point of contact under the foot, precise knowledge of the terrain’s slope in 3D at
 that point, and measurement of the reaction force along that tangent. To facilitate comparisons
 with experimental data, we consider an easier to measure correlate of ϵ_t via the parameter $\hat{\epsilon}_t$ that
 is defined as

$$276 \quad \hat{\epsilon}_t = 1 - \frac{\Delta v_x}{v_x}, \quad (5)$$

where $\Delta v_x/v_x$ is the fraction of the forward momentum of the runner lost due to the passive
 collision. On perfectly flat terrain, $\epsilon_t = \hat{\epsilon}_t$.

In the Monte Carlo simulations, $\hat{\epsilon}_t$ is characterized by a distribution that evolves with increasing
 steps (Fig. 4a, supplement Fig. S6a). The dependence of $\hat{\epsilon}_t$ on steps taken arises because the runner
 is slowing down, and thus v_x and consequently Δv_x change from step-to-step. But, the mean of
 $\hat{\epsilon}_t$ appears to converge to a constant after just 3 steps for all values of ϵ_{tc} (supplement Fig. S6b).
 Importantly, mean $\hat{\epsilon}_t$ increases linearly with ϵ_{tc} (Fig. 4b) and is thus a reliable correlate of the true
 tangential collision. However, $\hat{\epsilon}_t$ has a reduced range; mean $\hat{\epsilon}_t = 0.81$ at $\epsilon_{tc} = 0$, and mean $\hat{\epsilon}_t = 0.97$
 at $\epsilon_{tc} = 1$. The standard deviation of the distributions converges to a value between 0.05 and 0.1 by
 approximately 10 steps for most values of ϵ_{tc} except when $\epsilon_{tc} \rightarrow 1$ (Fig. 4a, supplement Fig. S6b).

For comparison, reported values of $\hat{\epsilon}_t$ from experiments with human runners on flat and two rough
 terrains are 0.94 ± 0.01 (mean \pm standard deviation) identically [22]. These experimental data are
 consistent with the prediction that optimal anticipatory runners should maintain $\epsilon_{tc} = 1$.

Figure 4: Estimated tangential coefficient of restitution $\hat{\epsilon}_t$ for anticipatory runners using Monte Carlo simulations with an ensemble size of 10^6 . **a**, Probability density function of $\hat{\epsilon}_t$ for human-like anticipatory runners with $\epsilon_{tc} = 1$ on rough terrain after 3 steps and after 20 steps. While the standard deviation almost doubles between the two distributions shown here (supplement Fig. S6c), the mean of the distribution converges by 3 steps (supplement Fig. S6b). **b**, Mean $\hat{\epsilon}_t$, converges by 3 steps for all values of ϵ_{tc} (supplement Fig. S6b), and is always less than 1, ranging from 0.81 at $\epsilon_{tc} = 0$ to 0.97 at $\epsilon_{tc} = 1$.

4 Linear stability analysis

For periodic dynamic systems, linear stability is defined as the response to small perturbations
 in the neighborhood of a periodic orbit [15, 23, 24] and analyzed using Floquet theory [15, 25].
 Floquet analysis for the stability of a periodic orbit defines a transverse cross-section to the orbit
 and a discrete return map from initial conditions on the cross-section back to the same cross-
 section after a complete period. The eigenvalues of the return map, called Floquet multipliers, are
 independent of the chosen cross-section and govern the stability of the periodic solution to small
 perturbations [25]. Here we consider the anticipatory runner and discuss the open-loop runner in
 supplement S6 because the unstable modes of both variants are the same.

The mechanical state of the runner is represented by $\zeta = (x, y, \phi, v_x, v_y, \omega)^T$, where (x, y) and ϕ
 denote the center of mass position and orientation, and (v_x, v_y) and ω are the respective velocities,
 all measured in a Newtonian reference frame that translates forward at a constant speed v_{x0} . A
 steady runner is periodic in this translating Newtonian frame of reference. We define a transverse
 cross-section (Poincaré section) at the apex of the aerial phase ($v_y = 0$) following the approach of
 Full et al. [23] and Seyfarth et al. [16]. The equations (1) yield the step-to-step return map \mathbf{f}_{an}

and its linearization \mathbf{T}_{an} in terms of the mechanical state $\boldsymbol{\psi}$ in a translating frame according to

$$306 \quad \boldsymbol{\psi} = (x, y, \phi, v_x, \omega)^{\text{T}}, \quad (6a)$$

$$307 \quad \boldsymbol{\psi}_{n+1} = \mathbf{f}_{\text{an}}(\boldsymbol{\psi}_n), \quad (6b)$$

$$308 \quad \Delta\boldsymbol{\psi}_{n+1} = \mathbf{T}_{\text{an}}\Delta\boldsymbol{\psi}_n, \quad (6c)$$

$$309 \quad \text{where } \Delta\boldsymbol{\psi} = \boldsymbol{\psi} - \boldsymbol{\psi}^*, \quad \mathbf{T}_{\text{an}} = \left. \frac{\partial \mathbf{f}_{\text{an}}}{\partial \boldsymbol{\psi}} \right|_{\boldsymbol{\psi}^*}. \quad (6d)$$

The Poincaré map given by equation (6b) has a fixed point at $\boldsymbol{\psi}^* = \mathbf{0}$ when the terrain is flat and
 corresponds to an exactly periodic runner on flat ground.

Figure 5: Illustration of the trajectory of the runner in state space in a reference frame that is translating along with the runner with velocity v_{x0} . The runner appears periodic in this reference frame and the runner's mechanical state follows a periodic orbit. The return map \mathbf{f}_{\bullet} (\bullet is 'ol' or 'an' for open-loop or anticipatory, respectively) is defined from the apex of the aerial phase ($v_y = 0$) to apex of the following aerial phase. $\boldsymbol{\psi}^*$ is the fixed point of the return map and $\boldsymbol{\psi}_n$ is a small perturbation away from the fixed point $\boldsymbol{\psi}^*$ at step n . In the next step, $\boldsymbol{\psi}_n$ maps to $\boldsymbol{\psi}_{n+1}$ at the apex of the following aerial phase under action of the return map \mathbf{f}_{\bullet} .

The linearized return map \mathbf{T}_{an} has three eigenvalues equal to one and the others are all less than
 one. The eigenvalues with magnitude less than one correspond to stable modes so that perturbations
 along their respective eigenvectors will always decay. The remaining three eigenvalues are all

$\lambda = 1$ with algebraic multiplicity equal to 3 and geometric multiplicity equal to 2. This implies
 that there are only two independent eigenvectors corresponding to the three unity eigenvalues
 and the matrix \mathbf{T}_{an} is therefore non-diagonalizable. For non-diagonalizable systems, the Jordan
 decomposition is used to analyze stability in terms of generalized eigenvectors (supplement S6),
 and implies that the modes (eigenvectors) associated with these eigenvalues cannot be decoupled
 and analyzed independently.

The two eigenvectors $\boldsymbol{\nu}_1$, $\boldsymbol{\nu}_2$ and the third generalized eigenvector $\boldsymbol{\nu}_3$ corresponding to the
 repeat eigenvalue $\lambda = 1$ span a subspace in which the dynamics of the return map ~~don't~~do not
 simply decay back to the origin. For a diagonalizable system, any perturbation within this subspace
 would neither decay nor grow. However, the non-diagonalizable nature of \mathbf{T}_{an} leads to the outcome
 that a perturbation $\Delta\boldsymbol{\psi}_0$ within this subspace grows with increasing steps. The eigenvectors $\boldsymbol{\nu}_1$,
 $\boldsymbol{\nu}_2$, $\boldsymbol{\nu}_3$, the initial perturbation $\Delta\boldsymbol{\psi}_0$, and its growth after n steps to $\Delta\boldsymbol{\psi}_n$ are given by,

$$327 \quad \boldsymbol{\nu}_1 = \begin{pmatrix} 0 & 0 & 1 & 0 & 0 \end{pmatrix}^T, \boldsymbol{\nu}_2 = \begin{pmatrix} 1 & 0 & 0 & 0 & 0 \end{pmatrix}^T, \boldsymbol{\nu}_3 = \begin{pmatrix} 0 & 0 & 0 & \frac{-1}{\sqrt{2}} & \frac{1}{\sqrt{2}} \end{pmatrix}^T, \quad (7a)$$

$$328 \quad \Delta\boldsymbol{\psi}_0 = \sum_{k=1}^3 \alpha_k \boldsymbol{\nu}_k, \text{ and} \quad (7b)$$

$$329 \quad \Delta\boldsymbol{\psi}_n = n\alpha_3\sqrt{2}\epsilon_n v_{y0}(\boldsymbol{\nu}_1 - \boldsymbol{\nu}_2) + \Delta\boldsymbol{\psi}_0, \text{ respectively.} \quad (7c)$$

As n grows larger, the asymptotic approximation (denoted by \approx) is given by

$$331 \quad \Delta\boldsymbol{\psi}_n \approx n\alpha_3\sqrt{2}\epsilon_n v_{y0} \begin{pmatrix} -1 \\ 0 \\ 1 \\ 0 \\ 0 \end{pmatrix} \text{ where } n \gg 1. \quad (8)$$

Only a perturbation of magnitude α_3 along $\boldsymbol{\nu}_3$ affects stability and leads to a nearly linear growth
 within the subspace spanned by the eigenvectors $\boldsymbol{\nu}_1, \boldsymbol{\nu}_2$. Perturbations along $\boldsymbol{\nu}_1$ or $\boldsymbol{\nu}_2$ neither
 grow nor decay because these represent invariance with respect to rotations and translations of
 the reference frame, respectively. A perturbation along the generalized eigenvector $\boldsymbol{\nu}_3$ may be
 geometrically viewed as one that conserves the velocity of the contact point on flat terrain but

changes the angular momentum of the runner about its center of mass. Therefore, any perturbation
 to the angular momentum will affect both orientation and forward speed.

For the special case of the anticipatory runner that completely avoids tangential collisions, the
 linearized return map \mathbf{T}_{an} with $\epsilon_{\text{tc}} = 1$ has eigenvalue $\lambda = 1$ of algebraic multiplicity 4 and geometric
 multiplicity 2, and one eigenvalue with $|\lambda| < 1$. The eigenvectors $\boldsymbol{\nu}_1, \boldsymbol{\nu}_2$ and the generalized
 eigenvectors $\boldsymbol{\nu}_3, \boldsymbol{\nu}_4$ associated with $\lambda = 1$ form a basis for a subspace within which an initial
 perturbation $\boldsymbol{\psi}_0$ grows linearly with the number of steps n in a subspace spanned by eigenvectors
 $\boldsymbol{\nu}_1, \boldsymbol{\nu}_2$, i.e.

$$345 \quad \boldsymbol{\nu}_1 = \begin{pmatrix} 0 \\ 0 \\ 1 \\ 0 \\ 0 \end{pmatrix}, \quad \boldsymbol{\nu}_2 = \begin{pmatrix} 1 \\ 0 \\ 0 \\ 0 \\ 0 \end{pmatrix}, \quad \boldsymbol{\nu}_3 = \begin{pmatrix} 0 \\ 0 \\ 0 \\ 0 \\ 1 \end{pmatrix}, \quad \boldsymbol{\nu}_4 = \begin{pmatrix} 0 \\ 0 \\ 0 \\ 1 \\ 0 \end{pmatrix}, \quad (9a)$$

$$346 \quad \Delta\boldsymbol{\psi}_0 = \sum_{k=1}^4 \alpha_k \boldsymbol{\nu}_k, \quad \Delta\boldsymbol{\psi}_n = 2\epsilon_n v_{y0} n (\alpha_3 \boldsymbol{\nu}_1 + \alpha_4 \boldsymbol{\nu}_2) + \Delta\boldsymbol{\psi}_0, \quad (9b)$$

$$347 \quad \Delta\boldsymbol{\psi}_n \approx n 2\epsilon_n v_{y0} \begin{pmatrix} \alpha_4 \\ 0 \\ \alpha_3 \\ 0 \\ 0 \end{pmatrix} \text{ for } n \gg 1. \quad (9c)$$

A perturbation to angular velocity ω causes a linear growth in orientation ϕ , and a perturbation to
 the linear velocity v_x causes a linear growth in position x . However, an anticipatory runner with
 $\epsilon_{\text{tc}} = 1$ avoids angular velocity perturbations due to the terrain altogether, i.e. $\alpha_3 = 0$. Therefore,
 only forward speed is affected due to the remaining unstable mode $\boldsymbol{\nu}_4$.

Although there are no unstable eigenvalues with magnitude greater than one, we find that
 the dynamics of running lead to an unstable growth with increasing steps. The growth due to
 non-diagonalizability of the return map is linearly proportional to the number of steps, and not
 geometric as is the case for simple unstable eigenvalues. Importantly, the primary effect of the

instability is to affect the forward speed and orientation, consistent with the numerical simulations
 that use finite perturbations and nonlinear dynamics. Also in agreement with the simulations, the
 only instability is translational when $\epsilon_{tc} = 1$.

**5 Scaling analysis of the orientational failure mode**

The mean steps to failure depends on many parameters, but none of the parameters separately
 predict the failure statistics (supplement Fig. S7). As most runners undergo orientational failures,
 we investigated whether the amount of body rotation accumulated over a single step due to a
 terrain slope perturbation would predict failure statistics.

If a runner with a periodic trajectory on flat ground encounters a sloped terrain of angle θ ,
 the orientation ϕ_{\bullet} at the next landing will no longer be vertical. This orientation ϕ_{\bullet} accumulated
 over one step depends on the take-off vertical velocity $v_{y,\bullet}^+$ via the aerial phase time $2v_{y,\bullet}^+$, and
 take-off angular velocity ω_{\bullet}^+ , as $\phi_{\bullet} = 2v_{y,\bullet}^+ \omega_{\bullet}^+$. The subscript ‘ \bullet ’ is a placeholder for ‘ol’ or ‘an’
 as the orientation change depends on whether the runner is purely open-loop (ol) or employs
 anticipatory (an) control. We hypothesize that the mean steps to failure N_{\bullet} is a function of
 the orientational threshold ϕ_{tol} and the orientation change over a single step ϕ_{\bullet} alone, i.e. $N_{\bullet} =$
 $s_{\bullet}(\phi_{tol}, \phi_{\bullet})$. Substituting the form of $s_{\bullet}(\phi_{tol}, \phi_{\bullet})$ derived in supplement S8, we show that the mean
 steps to failure N_{\bullet} is predicted to scale according to,

$$373 \quad N_{\bullet} \sim \frac{\phi_{tol}}{\phi_{\bullet}}, \quad (10)$$

where the expression for ϕ_{\bullet} is given in supplement equation S15e.

The mean steps to failure in simulations performed with many different parameter values (sup-
 plement S7) are well-approximated by a single function of a dimensionless parameter $\phi_{tol}/\phi_{\bullet}$
 (Fig. 6). The collapse of the simulation data highlights that the spin accumulated in one step
 due to a single perturbation (equation (10)) captures the fundamental principle underlying orien-
 tational failures. Importantly, this dimensionless parameter collapses the simulation data better
 than any individual parameter (supplement Fig. S7). Thus, the single parameter $\phi_{tol}/\phi_{\bullet}$ quantifies
 stability of runners of different sizes and mass distributions.

The dimensionless parameter $\phi_{tol}/\phi_{\bullet}$ also captures the parametric dependence of mean steps to

Figure 6: Generalizing results from section 3.1 and section 3.4 to a wider range of physical and terrain parameters. Mean steps to failure from the Monte Carlo simulations is plotted against **a**, $\phi_{\text{tol}}/\phi_{\text{an}}$ and **b**, $\phi_{\text{tol}}/\phi_{\text{ol}}$ for different values of ϕ_{tol} . The mean steps to failure depend mostly on a single dimensionless parameter $\phi_{\text{tol}}/\phi_{\bullet}$. All simulation parameters were varied independently in these simulations. But, for clarity, only variations in ϕ_{tol} are identified with different marker types.

failure on ϵ_{tc} and ϵ_{n} as seen from comparing the contour plots of $\phi_{\text{tol}}/\phi_{\bullet}$ shown in Fig. 7 against
 that of the direct simulations in Fig. 2d and Fig. 3a. The dependence of mean steps to failure on
 ϵ_{tc} and ϵ_{n} for small slopes of the terrain is understood using a series expansion of ϕ_{\bullet} in terms of θ
 as given by,

$$387 \quad \phi_{\text{ol}} = \left(\frac{2v_{y0}^2}{1 + I/G} \right) \theta + \left(\frac{3 - I/G}{(1 + I/G)^2} + \frac{4I/G}{(1 + I/G)^2} \epsilon_{\text{tc}} + \frac{2}{1 + I/G} \epsilon_{\text{n}} \right) v_{x0} v_{y0} \theta^2 + O(\theta^3), \quad (11a)$$

$$388 \quad \phi_{\text{an}} = \left(\frac{2v_{y0}^2}{1 + I/G} (1 - \epsilon_{\text{tc}}) \right) \theta + \left(\frac{3 - I/G}{(1 + I/G)^2} + \frac{5I/G - 3}{(1 + I/G)^2} \epsilon_{\text{tc}} - \frac{4I/G}{(1 + I/G)^2} \epsilon_{\text{tc}}^2 + \right. \\ 389 \quad \left. \frac{2(1 - \epsilon_{\text{tc}})}{1 + I/G} \epsilon_{\text{n}} \right) v_{x0} v_{y0} \theta^2 + O(\theta^3). \quad (11b)$$

For the open-loop strategy, neither of the collision parameters, ϵ_{n} or ϵ_{tc} , appear in the linear (leading
 order) term. When using an anticipatory strategy, the tangential collision parameter ϵ_{tc} appears
 to leading order. The normal collision parameter ϵ_{n} affects the second order dependence on θ for
 both strategies. These show why it is impossible to avoid orientational failures for the open-loop
 strategy, but may be avoided when using the anticipatory strategy by choosing $\epsilon_{\text{tc}} = 1$.

For the open-loop runner, $\phi_{\text{tol}}/\phi_{\text{ol}}$ is independent of ϵ_{n} and ϵ_{tc} to first order in θ (equation (11a)).
 Hence the contours in Fig. 7a (which resemble the contours in Fig. 2d from the Monte Carlo simu-
 lations) show a weak dependence on ϵ_{n} and ϵ_{tc} that arises from the θ^2 term in equation (11a). The
 parameter ϕ_{ol} is smallest when $\epsilon_{\text{n}} = \epsilon_{\text{tc}} = 0$, and largest when $\epsilon_{\text{n}} = \epsilon_{\text{tc}} = 1$. For a human-like run-
 ner, $I/G \ll 1$ (table 1), and thus the θ^2 term in equation (11a) can be reduced to $(3 + 2\epsilon_{\text{n}})v_{x0}v_{y0}\theta^2$,
 with no dependence on ϵ_{tc} at the asymptotic limit of $I/G \ll 1$. The asymptotic analysis of ϕ_{ol}

Figure 7: Contour plots of **a**, $\phi_{\text{tol}}/\phi_{\text{ol}}$ and **b**, $\phi_{\text{tol}}/\phi_{\text{an}}$ as a function of ϵ_n and ϵ_{tc} reveal that a single parameter captures the dependence of mean steps to failure of both open-loop runners (Fig. 2d) and anticipatory runners (Fig. 3a) on the collision parameters ϵ_n and ϵ_{tc} . Recall that the only controllable parameters for the runners in these simulations are ϵ_n and ϵ_{tc} . The complete expression for ϕ_{\bullet} , shown in supplement equation S15e, was used to generate these plots with parameter values drawn from table 1, and $\phi_{\text{tol}} = 1$. For the anticipatory runner, we restricted the maximum value of ϵ_{tc} to 0.99. For higher values of ϵ_{tc} , orientational failures do not dominate and thus mean failure statistics are not accounted for by $\phi_{\text{tol}}/\phi_{\bullet}$.

therefore explains why the contours of mean steps to failure in the Monte Carlo simulations are
 nearly parallel to the ϵ_{tc} axis and increase only slightly when ϵ_n is decreased (Fig. 2d).

For the anticipatory runner, the first order term in the expansion depends on ϵ_{tc} (equation 11b),
 unlike the case for the open-loop runner (equation (11a)). Nearly perfect anticipation corresponds
 to $\epsilon_{\text{tc}} \rightarrow 1$. At this limit $\phi_{\text{an}} \rightarrow 0$ and thus $N = \phi_{\text{tol}}/\phi_{\text{an}} \rightarrow \infty$, explaining why the contours of mean
 steps to failure in the Monte Carlo simulations are tightly bunched together in the neighborhood
 of $\epsilon_{\text{tc}} = 1$ (Fig. 3a) and nearly parallel to the ϵ_n axis. Like for the open-loop runner, ϕ_{an} also shows
 a dependence on ϵ_n only in the pre-factor to the θ^2 term of the series expansion in equation (11b).
 As ϵ_n decreases so does ϕ_{an} , and thus $\phi_{\text{tol}}/\phi_{\text{an}}$ increases, capturing the trend observed in the Monte
 Carlo simulations where decreasing ϵ_n increases steps taken for the anticipatory runner (Fig. 3a).
 For the anticipatory runner, unlike the open-loop runner, the ϵ_n dependence is coupled to ϵ_{tc} , and
 thus the sensitivity of the parameter $\phi_{\text{tol}}/\phi_{\bullet}$ to changes in ϵ_n depend on the value of ϵ_{tc} . The limit
 of $\epsilon_{\text{tc}} = 0$, where $\phi_{\text{an}} = \phi_{\text{ol}}$ has already been discussed above, for the open-loop runner. To analyze
 the case where $\epsilon_{\text{tc}} \rightarrow 1$, we approximate ϕ_{an} in the limit where $I/G \ll 1$ (e.g. human-like runners)
 as

$$416 \quad \phi_{\text{an}} \approx 2v_{y0}^2(1 - \epsilon_{\text{tc}})\theta + (1 - \epsilon_{\text{tc}})(2\epsilon_n + 3)v_{x0}v_{y0}\theta^2. \quad (12)$$

To understand the dependence of the mean steps to failure N on ϵ_n and ϵ_{tc} , we consider the
 limit of small angles of the terrain slope $\theta \ll 1$. Using equation (12), and for small θ we find that

mean steps to failure $N = \phi_{\text{tol}}/\phi_{\text{an}}$ and its sensitivity to changes in ϵ_n are given by

$$420 \quad N = \left(\frac{1}{1 - \epsilon_{\text{tc}}} \right) \frac{\phi_{\text{tol}}}{v_{y0}\theta} (2v_{y0} - 3v_{x0}\theta - 2v_{x0}\theta\epsilon_n), \quad (13a)$$

$$421 \quad \frac{\partial N}{\partial \epsilon_n} = - \left(\frac{1}{1 - \epsilon_{\text{tc}}} \right) \frac{2\phi_{\text{tol}}v_{x0}}{v_{y0}}. \quad (13b)$$

Therefore, N is more sensitive to changes in ϵ_n when $\epsilon_{\text{tc}} \rightarrow 1$. This resembles Fig. 3a where the
 mean steps to failure from the Monte Carlo simulations increases significantly as ϵ_n is reduced when
 $\epsilon_{\text{tc}} \rightarrow 1$, as opposed to when $\epsilon_{\text{tc}} \rightarrow 0$
[revised manuscript text omitted]

~~discussed how energy storage in the direction normal to the terrain (ϵ_n) and tangential collision~~
~~modulation (ϵ_{tc}) affects $\phi_{\text{tol}}/\phi_{\bullet}$ in section 5, and now turn to the implications the parameter has~~
~~on how body morphology and mass distribution affect stability.~~

The parameter ϕ_{tol} is the maximum angle of tilt the runner can accumulate before it falls. ~~By~~
~~employing~~ Animals that posses larger (base/height) ratios, i.e. ~~adopting~~ have a landscape rather
than a portrait orientation when viewed in the sagittal plane, ~~animals can~~ increase ϕ_{tol} and thereby
increase the parameter $\phi_{\text{tol}}/\phi_{\bullet}$. Quadrupeds such as cats and dogs, and other adept runners such
as cockroaches possess such an aspect ratio. ~~Another way to increase ϕ_{tol} is by increasing the~~
Larger range of motion of the leg with respect to the body ~~. Even if the body begins to tilt, the~~
~~ability to place the~~ also increases ϕ_{tol} as it improves the ability of the animal to correct for body
tilt by placing the foot in front of the runner ~~initiates stance and hence allows the runner to correct~~
~~for body orientation when initiating stance~~. In our simulations, the choice of ϕ_{tol} value is based on
this consideration of the leg angle for humans. ~~Ostriches are another example of an animal with a~~
~~portrait orientation but who are adept runners~~, perhaps in part due to the large of Other adept
runners with portrait orientations such as ostriches also posses a large range of motion of their legs.
Penguins, who are not known to be adept runners, occupy the ~~opposite end of~~ lower end of the
ϕ_{tol} scale due to possessing a portrait orientation ~~and as well as~~ low range of motion of their legs
compared to other bipeds such as humans, turkeys, and ostriches.

~~Because $\phi_{\text{tol}}/\phi_{\bullet} \propto I/G/v_{y0}^2$ (-), lowering~~ Low take-off angles, and
increased radius of gyration r_g relative to leg length r_l would be beneficial to stability as $\phi_{\text{tol}}/\phi_{\bullet} \propto I/G/v_{y0}^2$
(equations (11)). However, very low take-off angles increase the risk of tripping on rough terrains.
~~Altering body mass distribution to increase the~~ A larger radius of gyration r_g relative to leg length

~~r_g also reduces ϕ_\bullet and thereby increases stability. This can~~ could be achieved by increasing distal
 masses in appendages like arms and legs. However, increasing distal masses in the leg increases the
 metabolic cost of running [33] via increased energetic cost associated with swinging the leg [34, 35]
 and may also lead to potentially injurious collisions. ~~Alternatively, light legs with a bulky, extended~~
~~torso or large head, like in~~ Running animals like horses and bison ~~, also increases~~ have light legs,
but increase r_g . ~~Lastly, tails by possessing extended torsos and large heads. Tails~~ in animals like
 cats and lizards, ~~while used for active stabilization and in complex maneuvers like righting reflexes~~
~~[36, 38] which are used in active stabilization [36],~~ are yet another means to increase r_g , thereby
 reducing ϕ_\bullet . Anticipatory runners further benefit from setting $\epsilon_t \approx 1$ like observed in humans
 [22], thereby drastically reducing ϕ_\bullet as discussed in section 5. Thus, the dimensionless parameter
 $\phi_{\text{tol}}/\phi_\bullet$ is qualitatively consistent with the body morphology of adept and poor animal runners and
 we propose that it can be used as a design criteria for running robots.

7 Notation

Scalars are denoted by italic symbols (e.g. m for mass of the runner, I for the moment of inertia),
 vectors by bold, italic symbols (\mathbf{J}_{imp} for push-off impulse, \mathbf{v} for velocity), points or landmarks in
 capitalized non-italic symbols (such as center of mass G in Fig. 1) and capitalized, bold, non-italic
 symbols for matrices (such as return map matrix \mathbf{T}_{an}). Vectors associated with a point, such
 as velocity of center of mass G are written as \mathbf{v}_G , with the uppercase alphabet in the subscript
 specifying the point in the plane. Angular momentum vectors or moment of inertia variables are
 subscripted with ‘/A’ representing angular momentum or moment of inertia computed about point
 A , such as $I_{/G}$ representing moment of inertia about center of mass G . The \hat{x} component of the
 velocity vector of point A \mathbf{v}_A is denoted with a subscript ‘A,x’, e.g. tangential velocity of the contact
 point P is written as $v_{P,t}$. The symbols v_{x0}, v_{y0} denote the initial forward and vertical velocities of
 the runner at take-off. Variables just before collision with the terrain are denoted by the superscript
 ‘-’, after passive collision but before push-off by the superscript ‘c’, and just after push-off by the
 superscript ‘+’. For example, angular velocity before collision is ω^- , after passive collision is ω^c
 and just after push-off is ω^+ .

8 Data Accessibility

All detailed derivations are included in the electronic supplementary material. No other data were
generated in this research.

**9 Authors' Contribution**

596 MV conceived of the model. ND and MV ran the simulations. All authors were involved in
the analysis of the model; SM, MV and ND performed the linear stability analysis, MV and ND
performed the one-step analysis to capture mean statistics, MV, SM and ND did the analysis of
the steps to failure distributions. ND and MV wrote the manuscript, and all authors edited it.

**10 Acknowledgments and funding** Competing interests

We have no competing interests.

**11 Funding**

This work was funded by the Human Frontiers Science Program and the Wellcome/DBT India
Alliance.

**12 Competing interests**

~~We have no competing interests.~~

**References**

- [1] Daley MA, Usherwood JR, Felix G, Biewener AA. Running over rough terrain: guinea fowl
maintain dynamic stability despite a large unexpected change in substrate height. *Journal*
*of Experimental Biology*. 2006 Jan;209(1):171–187. Available from: [http://dx.doi.org/10.](http://dx.doi.org/10.1242/jeb.01986)
[1242/jeb.01986](http://dx.doi.org/10.1242/jeb.01986).
- [2] Birn-Jeffery AV, Daley MA. Birds achieve high robustness in uneven terrain through active
control of landing conditions. *Journal of Experimental Biology*. 2012 May;215(12):2117–2127.
Available from: <http://dx.doi.org/10.1242/jeb.065557>.
- [3] Grimmer S, Ernst M, Günther M, Blickhan R. Running on uneven ground: leg adjustment to
vertical steps and self-stability. *Journal of Experimental Biology*. 2008 Sep;211(18):2989–3000.
Available from: <http://dx.doi.org/10.1242/jeb.014357>.
- [4] Voloshina AS, Ferris DP. Biomechanics and energetics of running on uneven terrain. *Journal*
*of Experimental Biology*. 2015 Jan;218(5):711–719. Available from: [http://dx.doi.org/10.](http://dx.doi.org/10.1242/jeb.106518)
[1242/jeb.106518](http://dx.doi.org/10.1242/jeb.106518).
- [5] Daley MA, Usherwood JR. Two explanations for the compliant running paradox: reduced work
of bouncing viscera and increased stability in uneven terrain. *Biology Letters*. 2010;6(3):418–
421.

- [6] Blum Y, Vejdani HR, Birn-Jeffery AV, Hubicki CM, Hurst JW, Daley MA. Swing-leg trajectory
of running guinea fowl suggests task-level priority of force regulation rather than disturbance
rejection. PLoS ONE. 2014 Jun;9(6):e100399. Available from: [http://dx.doi.org/10.1371/
journal.pone.0100399](http://dx.doi.org/10.1371/journal.pone.0100399).
- [7] Karszen JD, Haberland M, Wisse M, Kim S. The effects of swing-leg retraction on running
performance: analysis, simulation, and experiment. *Robotica*. 2015 May;33(10):2137–2155.
Available from: <http://dx.doi.org/10.1017/S0263574714001167>.
- [8] Blickhan R. The spring-mass model for running and hopping. *Journal of Biomechanics*. 1989
Jan;22(11-12):1217–1227. Available from: [http://dx.doi.org/10.1016/0021-9290\(89\)
90224-8](http://dx.doi.org/10.1016/0021-9290(89)90224-8).
- [9] McMahon TA, Cheng GC. The mechanics of running: how does stiffness couple with speed?
*Journal of Biomechanics*. 1990 Jan;23:65–78. Available from: [http://dx.doi.org/10.1016/
0021-9290\(90\)90042-2](http://dx.doi.org/10.1016/0021-9290(90)90042-2).
- [10] Blickhan R, Full R. Similarity in multilegged locomotion: bouncing like a monopode. *Journal*
*of Comparative Physiology A*. 1993 Nov;173(5):509–517. Available from: [http://dx.doi.
org/10.1007/BF00197760](http://dx.doi.org/10.1007/BF00197760).
- [11] Srinivasan M, Holmes P. How well can spring-mass-like telescoping leg models fit multi-
pedal sagittal-plane locomotion data? *Journal of Theoretical Biology*. 2008 Nov;255(1):1–7.
Available from: <http://dx.doi.org/10.1016/j.jtbi.2008.06.034>.
- [12] Pearson K. Common principles of motor control in vertebrates and invertebrates. *Annual*
*Review of Neuroscience*. 1993 Mar;16(1):265–297. Available from: [http://dx.doi.org/10.
1146/annurev.ne.16.030193.001405](http://dx.doi.org/10.1146/annurev.ne.16.030193.001405).
- [13] Pearson KG. Proprioceptive regulation of locomotion. *Current Opinion in Neurobiology*. 1995
Dec;5(6):786–791. Available from: [http://dx.doi.org/10.1016/0959-4388\(95\)80107-3](http://dx.doi.org/10.1016/0959-4388(95)80107-3).
- [14] Dickinson MH, Farley CT, Full RJ, Koehl M, Kram R, Lehman S. How animals move: an
integrative view. *Science*. 2000 Apr;288(5463):100–106. Available from: [http://dx.doi.org/
10.1126/science.288.5463.100](http://dx.doi.org/10.1126/science.288.5463.100).
- [15] Holmes P, Full RJ, Koditschek D, Guckenheimer J. The dynamics of legged locomotion:
Models, analyses, and challenges. *SIAM Review*. 2006 Jan;48(2):207–304. Available from:
<http://dx.doi.org/10.1137/S0036144504445133>.
- [16] Seyfarth A, Geyer H, Herr H. Swing-leg retraction: a simple control model for stable running.
*Journal of Experimental Biology*. 2003 Aug;206(15):2547–2555. Available from: [http://dx.
doi.org/10.1242/jeb.00463](http://dx.doi.org/10.1242/jeb.00463).
- [17] Cavagna G, Saibene F, Margaria R. Mechanical work in running. *Journal of Applied Physi-*
*ology*. 1964 Mar;19(2):249–256. Available from: [http://dx.doi.org/10.1152/jappl.1964.
19.2.249](http://dx.doi.org/10.1152/jappl.1964.19.2.249).
- [18] van Beers RJ, Baraduc P, Wolpert DM. Role of uncertainty in sensorimotor control. *Philosoph-*
*ical Transactions of the Royal Society B: Biological Sciences*. 2002 Aug;357(1424):1137–1145.
Available from: <http://dx.doi.org/10.1098/rstb.2002.1101>.

- [19] Voloshina AS, Kuo AD, Daley MA, Ferris DP. Biomechanics and energetics of walking on
uneven terrain. *Journal of Experimental Biology*. 2013 Aug;216(21):3963–3970. Available
from: <http://dx.doi.org/10.1242/jeb.081711>.
- [20] Byl K, Tedrake R. Metastable walking machines. *The International Journal of Robotics*
*Research*. 2009 Jun;28(8):1040–1064. Available from: [http://dx.doi.org/10.1177/](http://dx.doi.org/10.1177/0278364909340446)
[0278364909340446](http://dx.doi.org/10.1177/0278364909340446).
- [21] Srinivasan M, Ruina A. Computer optimization of a minimal biped model discovers walking
and running. *Nature*. 2006 Sep;439(7072):72–75. Available from: [http://dx.doi.org/10.](http://dx.doi.org/10.1038/nature04113)
[1038/nature04113](http://dx.doi.org/10.1038/nature04113).
- [22] Dhawale N, Venkadesan M. Energetics and stability of running on rough terrains. *American*
*Society of Biomechanics*; 2018. .
- [23] Full RJ, Kubow T, Schmitt J, Holmes P, Koditschek D. Quantifying dynamic stability and ma-
neuverability in legged locomotion. *Integrative and Comparative Biology*. 2002 Feb;42(1):149–
157. Available from: <http://dx.doi.org/10.1093/icb/42.1.149>.
- [24] Bruijn S, Meijer O, Beek P, Van Dieën J. Assessing the stability of human locomotion: a
review of current measures. *Journal of the Royal Society Interface*. 2013 Mar;10(83):20120999.
Available from: <http://dx.doi.org/10.1098/rsif.2012.0999>.
- [25] Guckenheimer J, Holmes P. Nonlinear oscillations, dynamical systems, and bifurcations
of vector fields. Springer-Verlag; 1983. Available from: [http://dx.doi.org/10.1007/](http://dx.doi.org/10.1007/978-1-4612-1140-2)
[978-1-4612-1140-2](http://dx.doi.org/10.1007/978-1-4612-1140-2).
- [26] Venkadesan M, Mandre S, Bandi MM. In: Sharbafi MA, Seyfarth A, editors. *Bioinspired*
*Legged Locomotion: Models, Concepts, Control and Applications*, Chapter 7. Butterworth-
Heinemann; 2017. .
- [27] Raibert M, Blankespoor K, Nelson G, Playter R, Team T. Bigdog, the rough-terrain quadruped
robot. *IFAC Proceedings Volumes*. 2008;41(2):10822–10825. Available from: [http://dx.doi.](http://dx.doi.org/10.3182/20080706-5-KR-1001.01833)
[org/10.3182/20080706-5-KR-1001.01833](http://dx.doi.org/10.3182/20080706-5-KR-1001.01833).
- [28] Blum Y, Lipfert S, Rummel J, Seyfarth A. Swing leg control in human running. *Bioinspira-*
*tion & Biomimetics*. 2010 May;5(2):026006. Available from: [http://dx.doi.org/10.1088/](http://dx.doi.org/10.1088/1748-3182/5/2/026006)
[1748-3182/5/2/026006](http://dx.doi.org/10.1088/1748-3182/5/2/026006).
- [29] Birn-Jeffery AV, Hubicki CM, Blum Y, Renjewski D, Hurst JW, Daley MA. Don't break a
leg: running birds from quail to ostrich prioritise leg safety and economy on uneven terrain.
*Journal of Experimental Biology*. 2014 Oct;217(21):3786–3796. Available from: [http://dx.](http://dx.doi.org/10.1242/jeb.102640)
[doi.org/10.1242/jeb.102640](http://dx.doi.org/10.1242/jeb.102640).
- [30] Kuo AD. Stabilization of lateral motion in passive dynamic walking. *The International Journal*
*of Robotics Research*. 1999 Sep;18(9):917–930. Available from: [http://dx.doi.org/10.1177/](http://dx.doi.org/10.1177/02783649922066655)
[02783649922066655](http://dx.doi.org/10.1177/02783649922066655).
- [31] Donelan JM, Kram R, et al. Mechanical and metabolic determinants of the preferred step
width in human walking. *Proceedings of the Royal Society of London B: Biological Sciences*.
2001;268(1480):1985–1992.

- [32] Arellano CJ, Kram R. The energetic cost of maintaining lateral balance during human running.
Journal of Applied Physiology. 2012 Feb;112(3):427–434. Available from: <http://dx.doi.org/10.1152/jappphysiol.00554.2011>.
- [33] Myers M, Steudel K. Effect of limb mass and its distribution on the energetic cost of running.
Journal of Experimental Biology. 1985;116(1):363–373.
- [34] Marsh RL, Ellerby DJ, Carr JA, Henry HT, Buchanan CI. Partitioning the energetics of
walking and running: swinging the limbs is expensive. Science. 2004 Jan;303(5654):80–83.
Available from: <http://dx.doi.org/10.1126/science.1090704>.
- [35] Doke J, Donelan JM, Kuo AD. Mechanics and energetics of swinging the human leg. Journal
of Experimental Biology. 2005 Feb;208(3):439–445. Available from: <http://dx.doi.org/10.1242/jeb.01408>.
- [36] Libby T, Moore TY, Chang-Siu E, Li D, Cohen DJ, Jusufi A, et al. Tail-assisted pitch
control in lizards, robots and dinosaurs. Nature. 2012 Jan;481(7380):181. Available from:
<http://dx.doi.org/10.1038/nature10710>.
- [37] ~~Arellano, C. J. and Kram, R. (2011). The effects of step width and arm swing on energetic
cost and lateral balance during running. *Journal of Biomechanics* 44, 1291–1295.~~
- [38] ~~Jusufi, A., Zeng, Y., Full, R. J. and Dudley, R. (2011). Aerial righting reflexes in
flightless animals.~~